# Rapid functional impairment of natural killer cells following tumor entry limits anti-tumor immunity

Isaac Dean [1], Colin Y. C. Lee [2,3], Zewen K. Tuong [2,3], Zhi Li [1], Christopher A. Tibbitt[4], Claire Willis[1], Fabrina Gaspal[1], Bethany C. Kennedy [1], Veronika Matei-Rascu[1], Rémi Fiancette [1], Caroline Nordenvall[5], Ulrik Lindforss[5], Syed Murtuza Baker [6], Christian Stockmann[7], Veronika Sexl[8], Scott A. Hammond [9], Simon J. Dovedi[10], Jenny Mjösberg [4,11], Matthew R. Hepworth [12], Gianluca Carlesso[9], Menna R. Clatworthy [2,3] ✉ & David R. Withers [1] ✉

Immune cell dysfunction within the tumor microenvironment (TME) undermines the control of cancer progression. Established tumors contain phenotypically distinct, tumor-specific natural killer (NK) cells; however, the temporal dynamics, mechanistic underpinning and functional significance of the NK cell compartment remains incompletely understood. Here, we use photo-labeling, combined with longitudinal transcriptomic and cellular analyses, to interrogate the fate of intratumoral NK cells. We reveal that NK cells rapidly lose effector functions and adopt a distinct phenotypic state with features associated with tissue residency. NK cell depletion from established tumors did not alter tumor growth, indicating that intratumoral NK cells cease to actively contribute to anti-tumor responses. IL-15 administration prevented loss of function and improved tumor control, generating intratumoral NK cells with both tissue-residency characteristics and enhanced effector function. Collectively, our data reveals the fate of NK cells after recruitment into tumors and provides insight into how their function may be revived.

The clinical impact of immunotherapy for cancer patients has prompted extensive research into the composition and function of immune cells within tumors[1-4]. Amongst these, tumor cytotoxic CD8 T cells have been the most extensively characterized, with a spectrum of exhausted T cells with diminishing effector functions described[5-8]. Innate lymphocytes, classically represented by NK cells, but which now also includes other innate lymphoid cell (ILC) populations, also have potent cytotoxic potential. There is increasing interest in harnessing

[1]Institute of Immunology and Immunotherapy, College of Medical and Dental Sciences, University of Birmingham, Birmingham, UK. [2]Department of Medicine, Molecular Immunity Unit, Medical Research Council Laboratory of Molecular Biology, University of Cambridge, Cambridge, UK. [3]Cellular Genetics, Wellcome Sanger Institute, Wellcome Genome Campus, Hinxton, Cambridge, UK. [4]Center for Infectious Medicine, Department of Medicine Huddinge, Karolinska Institutet, Stockholm, Sweden. [5]Department of Molecular Medicine and Surgery, Karolinska Institutet and Department of Pelvic Cancer, Karolinska University Hospital, Stockholm, Sweden. [6]Division of Informatics, Imaging & Data Science, Faculty of Biology, Medicine and Health, the University of Manchester, Manchester Academic Health Science Centre, Manchester, UK. [7]Institute of Anatomy, University of Zurich, Zurich, Austria. [8]Institute of Pharmacology and Toxicology, University of Veterinary Medicine, Vienna, Austria. [9]Early Oncology R&D, AstraZeneca, MD, USA. [10]Early Oncology R&D, AstraZeneca, UK. [11]Clinical Lung and Allergy Research, Medical unit for Lung and Allergy Diseases, Karolinska University Hospital, Stockholm, Sweden. [12]Lydia Becker Institute of Immunology and Inflammation, Division of Infection, Immunity and Respiratory Medicine, Faculty of Biology, Medicine and Health, the University of Manchester, Manchester Academic Health Science Centre, Manchester, UK. ✉e-mail: mrc38@medschl.cam.ac.uk; d.withers@bham.ac.uk

this expanded pool of effector cells for therapeutic purposes by unlocking its cytotoxic potential. However, the behavior of these cells and how this changes over time within the TME remains poorly understood.

Conventional NK (cNK) cells are circulatory in mouse and human with a robustly cytotoxic phenotype; they are armed with an array of granzymes and perforin, alongside multiple apoptosis-inducing ligands including TRAIL and FASL[9–11]. In vivo, constitutive NK cell deficiency results in impaired control of tumor growth and enhanced metastatic disease in multiple murine cancer models[10,12]. Importantly, beyond their direct cytotoxic functions, NK cells further act as orchestrators of the T cell response through their recruitment, activation and expansion of dendritic cells (DCs), particularly conventional type 1 DCs (cDC1), via their production of the chemokines CCL5, XCL1, as well as FLT3L and IFNγ[11,13,14]. The cDC1 subset is the most adept at trafficking to the draining lymph node (dLN) to cross-present tumor antigens to naïve T cells, and thus drive the expansion of anti-tumor CD8 T cells[15,16]. Furthermore, cDC1 within the TME attract and restimulate effector CD8 T cells to amplify and sustain the response[17,18]. Collectively, these data emphasize the key role of coordinated NK cell-DC crosstalk for generating durable anti-tumor responses. Indeed, the frequency of NK and cDC1 within melanoma is predictive of the response to anti-PD-1 therapy[14].

However, phenotypic heterogeneity and dysfunction has been described within the intratumoral NK cell compartment[19,20], likely undermining control of tumor growth. The discovery of different ILC populations involved in type 1 immune responses, collectively termed ILC1s, adds a further level of complexity to understanding the composition of tumor infiltrating lymphocytes (TILs)—most notably whether such populations derive from cells recruited from the circulation or tissue-resident compartments[21–24]. ILC1s share many phenotypic similarities and transcriptional programs with NK cells, and although most ILCs are tissue-resident, circulating ILC1s have been observed in mice[25–29]. NK cells are developmentally distinct from ILC1s[25,30,31] and were initially further distinguished based upon granzyme expression and cytotoxicity, however, more recent descriptions have revealed that some ILC1s express granzymes and are able to kill target cells[24,32,33]. Indeed, NK cell conversion to an ILC1 state within the TME has been proposed[19]. Thus, NK cell dysfunction within tumors may reflect the adaptation of cNKs to the TME, the contribution of local ILC populations, or a combination of both. Defining how and why intratumoral NK/ILC1 populations form may facilitate more precise investigation of the mechanisms that need to be targeted to manipulate these innate lymphocytes and promote anti-tumor immune responses.

Here, we use temporal labeling of tumors through photoconversion[34] to track the fate of NK cells in vivo after recruitment into solid tumors. Our data reveals the rapid loss of chemokine and cytokine production, alongside impaired cytotoxicity, as cNK cells adapt to, or are modulated by, the TME. We demonstrate that all cNK cells retained within the tumor ultimately adopt a distinct, dysfunctional state, characterized by expression of CD49a, and that the heterogeneity observed across many pre-clinical tumor models reflects the time cNK cells have spent within the tumor. Depletion of NK cells from established tumors had no impact on tumor growth indicating that these cells have ceased to actively contribute to tumor control. The loss of NK cell functionality after tumor entry could be blocked through enhanced IL-15 signaling, leading to improved control of tumor growth. Collectively, our data clarify the fate of NK cells within solid tumors, defining the tumor-adapted state that these cells adopt once within the TME, which limits their contribution to the anti-tumor response. These data inform further efforts to revive intratumoral NK cells and enhance anti-tumor immunity.

## Results

### NK cells rapidly exhibit transcriptome alterations after tumor entry

Recent studies have highlighted that the intratumoral NK cell compartment is distinct, characterized by altered functions including reduced cytotoxicity and an impaired ability to orchestrate DC recruitment and activation[19–21,33]. Precisely why this occurs remains unresolved. To date, in vivo studies tracking the fate of NK cells specifically within the tumor are lacking. Using our recently described dynamic labeling of the tumor immune compartment[34], we sought to define the state of NK cells as they enter tumors and map real-time changes in their phenotype and function over time. To this end, MC38 tumors were grafted subcutaneously on the flank of Kaede photoconvertible mice, a transgenic model in which all cells express a green fluorescent protein that irreversibly switches to a red form upon exposure to violet light[35]. The entire immune compartment of the tumor was selectively labeled[34] at 13 days post-engraftment and tissue harvested 2 days later. Thus, amongst the cells isolated from the tumor, unlabeled Kaede Green+ (KG+) cells had been in the tumor for up to 48 h, while Kaede Red+ (KR+) cells had spent at least 48 h within the tissue. To capture transcriptomic changes across the TIL compartment over time and in an unbiased manner, we employed scRNA-sequencing (scRNA-seq). TILs were FACS-isolated to better equilibrate numbers, further split into KG+ and KR+ populations and analyzed by droplet-based scRNA-seq (Supplementary Fig. 1A). The gating strategy used ensured the NK cell compartment was fully captured (Supplementary Fig. 1B).

After quality control, a total of 46,342 TILs were analyzed, comprised of a small number of B cells, putative NK cells and multiple T cell populations (CD4, CD8, Treg, γδ TCR+) as defined by canonical markers (Supplementary Fig. 1C−E). We focused upon the putative NK cell cluster of 11,808 cells, initially defined by high expression of *Prf1*, *Ncr1*, *Klrb1c* and *Fcgr3*. Unbiased re-clustering of these cells in isolation revealed 8 clusters, of which 6 were defined as NK cells based upon expression of *Ncr1*, *Eomes*, *Gzmb* and the absence of *Cd3e* and *Il7ra*, including a cycling NK cell cluster additionally expressing *Mki67* and *Birc5* (Fig. 1A, B, Supplementary Fig. 1F). The remaining two clusters were identified as NKT cells and an ILC cluster, distinguished from bona fide NK cells by expression of *Il7ra*, *Rora* and *Gata3*, in the absence of *Eomes*, and *Cd3e* expression (Supplementary Fig. 1G−I)[31,36]. Expression of *Tbx21*, but not *Eomes* within the small ILC cluster indicated the prevalence of ILC1s (Supplementary Fig. 1I)[31]. Using published NK and ILC1-specific signatures[21,23,37], we further confirmed our cluster identification (Supplementary Fig. 1J).

We then determined the proportion of KG+ and KR+ cells within each cluster to contextualize the time spent within the tumor microenvironment of each sub-cluster (Fig. 1C−E). These data revealed a gradation in the proportion of KR+ cells across the NK clusters, with the NK_1 cluster comprised almost entirely of KG+ cells and the NK_5 cluster at the other extreme, consisting of approximately 90% KR+ cells. Further analysis of the NK_cycling cluster revealed that cycling cells could be found in all NK/ILC/NKT cell clusters, suggesting that these do not constitute a distinct phenotype (Supplementary Fig. 1K). However, of note, NK_5 was particularly enriched among the cycling NK cells, suggesting that some tumor-retained NK cells may expand in situ (Supplementary Fig. 1L). To temporally order the transcriptomic changes that occur within NK cells after entering the tumor we performed a pseudotime trajectory analysis, rooted in NK_1 since this contained the highest proportion of NK cells that had recently entered the tumor. These analyses highlighted the NK_5 cluster as the end state of the trajectory (Fig. 1F, G). Similarly, partition-based graph abstraction (PAGA) to map cluster connectivity (Fig. 1H), indicated that the major cell fate trajectory for tumor NK cells progresses from the NK_1 through to NK_2 and then onto the NK_5 transcriptional signature.

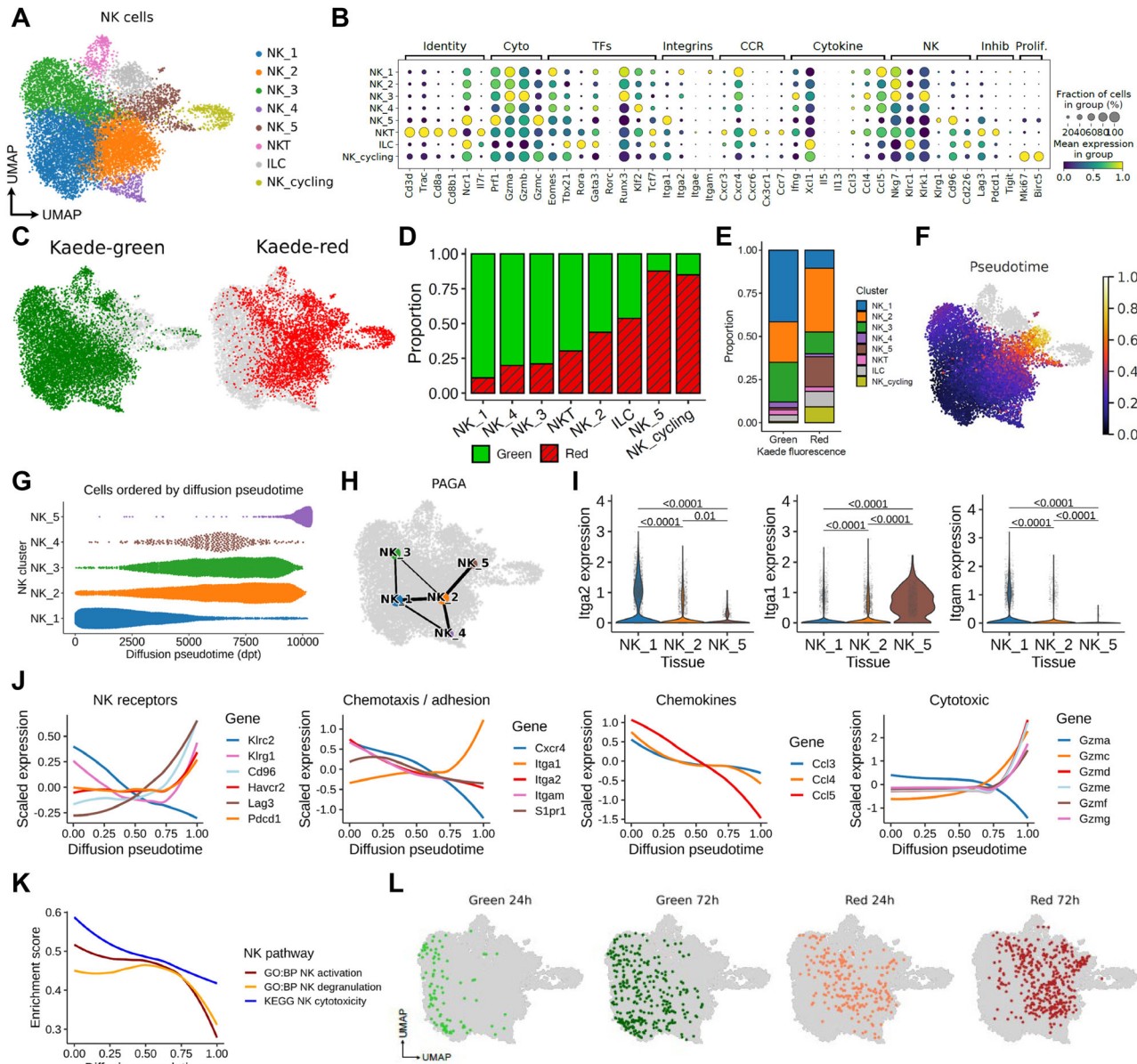

**Fig. 1 | Rapid changes to the NK cell transcriptome after entry into the TME.** MC38 tumors, grafted subcutaneously on the flank, were photoconverted and analyzed 48 h later using droplet-based scRNA-sequencing. **A** UMAP showing 11,808 NK cells defined by expression of *Ncr1*, *Prf1*, *Klrb1c*, *Fcgr3*, resolved into 8 clusters comprised of NK_1 to NK_5, alongside 1 cycling cluster and two further clusters that describe ILC and NKT cells. **B** Dot plots showing expression of selected genes used to further characterize the clusters. **C** UMAP showing the distribution of Kaede Green+ and Kaede Red+ cells across the NK clusters. **D** The proportion of Kaede Green+ and Kaede Red+ cells within each cluster. **E** Proportion of each cluster within the Kaede Green+ and Kaede Red+ cells. **F** UMAP showing diffusion pseudotime trajectory rooted in NK_1. **G** Bee swarm plot of the 5 NK clusters (NK_1 to NK_5) over pseudotime. **H** PAGA illustrating gene expression relationship between NK clusters. **I** Violin plots showing expression of *Iga1*, *Itga2*, *Itgam* across NK_1, NK_2 and NK_5. **J** Differentially expressed genes over pseudotime grouped by function. **K** Pathway analyses characterizing changes in NK activation, degranulation, and cytotoxicity over pseudotime. **L** UMAPs showing integration of NK cell data after 24 and 72 h post-photoconversion with the original data derived from 48 h post-labeling. Statistical significance determined by two-sided Wilcoxon rank-sum test with Benjamini-Hochberg multiple-testing correction. Data are shown as box (median; box, 25th percentile and 75th percentile; whiskers, 1.5*inter-quartile range) and violin plots. In (**J**, **K**) data are shown as local regression (loess) fit to scaled expression values.

Differential expression of the integrins CD49a and CD49b has been used to distinguish cNK cells (CD49b+) and tissue-resident NK cells and ILC1 (CD49a+)[21]. Expression of *Itga2* (CD49b) was limited to the NK_1 and NK_2 clusters while *Itga1* (CD49a) was expressed by the majority of the NK_5 cluster, but not the other clusters (Fig. 1I). Interestingly, expression of *Itgam*, which encodes CD11b and defines mature NK cells in the circulation[38], was limited to the NK_1 cluster.

To begin investigating the transcriptomic changes occurring within intratumoral NK cells, we plotted the expression of functionally grouped genes over pseudotime (Fig. 1J)[39]. As NK cells progressed towards the terminal intratumoral pseudotime state (NK_5), there was an upregulation of inhibitory receptors including *Pdcd1*, *Lag3*, *Havcr2*, *Cd96*, a change in migration-associated transcripts with a marked reduction in *Cxcr4* and *Itgam* expression, but an increase in *Itga1*, as well as loss of expression of *Ccl3*, *Ccl4*, *Ccl5* (chemokines associated with DC recruitment), and altered granzyme expression (Fig. 1J). Consistent with these findings, geneset enrichment analysis identified a reduction in 'NK activation', 'NK degranulation' and 'NK cytotoxicity' pathway gene sets over pseudotime (Fig. 1K). To further investigate the potential drivers of this transition, we performed a gene regulatory

network analysis using SCENIC[40] (Supplementary Fig. 2). This revealed the concomitant downregulation of a large number of transcription factor regulons over pseudotime, as NK cells transition to the state described by NK_5. These included the AP-1 complex transcription factors *Fos, Fosb, Jun, Junb*, which are activated during NK cell cytolytic programs and down regulated by interactions with inhibitory ligands[41]. Other down-regulated transcription factors included *Irf8, Klf2, Myc*, which support NK cell activation and proliferation[42–44]. Interestingly, we did not identify transcription factor regulons significantly upregulated with pseudotime, suggesting that the tumor-retained NK state arises from the reduced activity of core transcription factors associated with promoting mature NK cell development and expansion.

To better understand changes in intratumoral NK cells over time, we additionally analyzed NK cells in scRNA-seq data where TILs were assessed at 24 and 72 h post photoconversion[34], providing finer granularity of the changes that occur over real time. Re-analysis of 1035 NK cells revealed three NK clusters (Supplementary Fig. 3A, B). Notably, the NK_a cluster was comprised almost entirely of KG+ cells, while NK_c contained only KR+ cells, consistent with a simple linear trajectory (Supplementary Fig. 3B, C). Substantial changes in gene expression across these three clusters closely matched those observed in our initial analysis, including a switch from *Itgam* to *Itga1* expression (Supplementary Fig. 3D, E). To determine where NK cell transcriptomes from 24 and 72 h post photoconversion would embed along the pseudo-time trajectory, we projected these data into our initial reference single cell data set (Fig. 1F). Reassuringly, these time-stamped data closely aligned to our pseudotime trajectory, validating that our analyses faithfully model temporal transcriptional changes (Fig. 1L). Finally, to further investigate the nature of the NK cells recruited into the tumor, we integrated circulating NK cells utilizing publicly available scRNA-seq date (https://www.10xgenomics.com/resources/datasets/10-k-mouse-pbm-cs-multiplexed-2-cm-os-3-1-standard-6-0-0) with our tumor data set (Fig. S1M). These blood NK cells most resembled NK_1 and NK_2, which are predominantly Kaede-green in the scRNA-seq data, and importantly, none were mapped to NK_5 (Supplementary Fig. 1N). Comparison of *Itgam, Itga1* and *Itga2* expression confirmed the absence of Itga1 amongst the circulating NK cells (Supplementary Fig. 1O).

Collectively, these data reveal rapid changes to the transcriptome of *Itgam*-expressing cNK cells after entering the TME from the circulation. Once within tumors, NK cells rapidly differentiate towards a common transcriptional state characterized by substantially altered core functions and expression of *Itga1*.

### Differential expression of CD11b and CD49a expression capture temporal changes in NK cell phenotype

To validate the rapid changes observed in the transcriptional profile of NK cells entering the tumor, we turned to flow cytometry and initially sought to establish how best to identify cells described along the main cell fate trajectory. Given the differential expression of *Itga1* and *Itga2* (Fig. 1I) and prior use of these integrins to define NK populations[19], we assessed CD49a versus CD49b on CD3- NK1.1+ cells in MC38 tumors. However, our analysis revealed only two clear NK cell populations (Fig. 2A), prompting us to try alternative gating strategies. Comparison of CD49a versus CD11b expression identified three populations of CD3- NK1.1+ cells: CD11b+ CD49a- cells, CD11b- CD49a- and CD11b- CD49a+ cells (Fig. 2B). Importantly, a similar staining pattern was observed across multiple murine cancer models including other subcutaneous (CT26, B16-F10-OVA), orthotopic (E0771) and primary (MMTV-PyMT) tumors (Fig. 2C, D). To extend this analysis and compare intratumoral NK cells with the NK/ILC1 populations found in healthy tissue, we assessed the CD3- NK1.1+ cells in MC38 and B16-F10 tumors, alongside spleen, liver, small intestine and colon (Supplementary Fig. 4). These data indicated that the tumor NK cell compartment was phenotypically distinct to the ILC1 populations identified in non-tumor tissues, consistent with recent scRNA-seq analyses[21].

To confirm that expression of CD49a versus CD11b could be used to assess the phenotype of NK cells as they enter the tumor and then track changes over time, we photoconverted MC38 tumors and analyzed the intratumoral NK cell compartment 5, 24 and 72 h later. Enumeration of the total number of NK cells indicated the continuous recruitment of new (KG+) cells over this time frame, alongside a modest decline in the number of retained (KR+) cells within the tumor (Fig. 2E). Analysis at only 5 h post-photoconversion revealed a small KG+ population of newly arrived NK cells, the majority of which were CD11b+CD49a- and the CD49a+ compartment was absent (Fig. 2F). Analysis at 24 h post-photoconversion revealed that approximately 10% of the KG+ NK cells expressed CD49a, establishing that a switch in integrin expression begins to occur after only 1 day in the TME. Notably, the KR+ population at 72 h post photoconversion, which by definition had spent at least 3 days within the tumor, were essentially all CD49a+ CD11b-. Since a mixture of CD49a+ and CD49a- NK cells were evident amongst the KR+ cells at 24 h, these data indicate that full conversion of cNKs takes more than 24 h but is completed within approximately 3 days. Analysis of the % of KG+ cells within each population highlighted that by 72 h post photoconversion, the vast majority of the CD11b+ CD49a- and CD11b- CD49a- populations comprised of cells entering post-labeling (Fig. 2G). Comparable dynamic changes in NK cell phenotype over time were observed in multiple other murine tumor models (Supplementary Fig. 5). In addition, we further validated the expression patterns observed in our transcriptomic analysis, identifying increased expression of LAG-3 and CD69 as characteristic features of the CD49a+ 'tumor -retained' state (Fig. 2H–J). Interestingly, expression of NKG2A, the inhibitory receptor encoded by *Klrc1*, showed a biphasic expression pattern, largely independent of time within the tumor.

These data clearly indicated that intratumoral NK cell heterogeneity could be explained by the time cNK cells have spent within the tumor. The absence of CD49a- NK cells amongst the KR+ NK cells at 72 h post-photoconversion indicates that these cells must all either differentiate to a CD49a+ state, egress the tumor, or die in situ. To investigate NK cell egress, tumors were photoconverted and the KR+ cells present in the dLN and spleen analyzed 24 and 72 h later (Supplementary Fig. 6). The proportion and total number of KR+ NK cells in either tissue was very low, indicating that few NK cells egressed the tumor. Of those NK cells that did egress, the majority lacked CD49a expression. Collectively these data indicate that the CD49a+ NK cell compartment in these murine tumor models arise from the differentiation and retention of circulating cNK cells.

### NK cells rapidly lose core effector functions after tumor entry

Having established how to identify the temporal changes in NK cell state through a combination of photo-labeling and analysis of integrin expression, we sought to understand how NK cell function was impacted by time spent in the tumor microenvironment. To consider the potential cellular interactions of the different NK cell states identified in an unbiased manner, we used CellChat[45] and specifically focused on myeloid cells within the TME given evidence of this cellular axis impacting anti-tumor immunity (Supplementary Fig. 7). This analysis identified a number of potential associations that differed depending on the NK state, including a number of immunostimulatory interactions (*Ifng, Ccl5, Il18*) involving newly recruited NK cells, but not those that had been retained within the tumor for several days. Other associations of note included the potential responsiveness of the NK_5 state to *Il15* signaling as well as the emergence of inhibitory receptor interactions (e.g., *Cd244a:Cd48*). Collectively, this unbiased in silico analysis highlighted the potential for quite different cellular communications between recently recruited and tumor-retained NK cells.

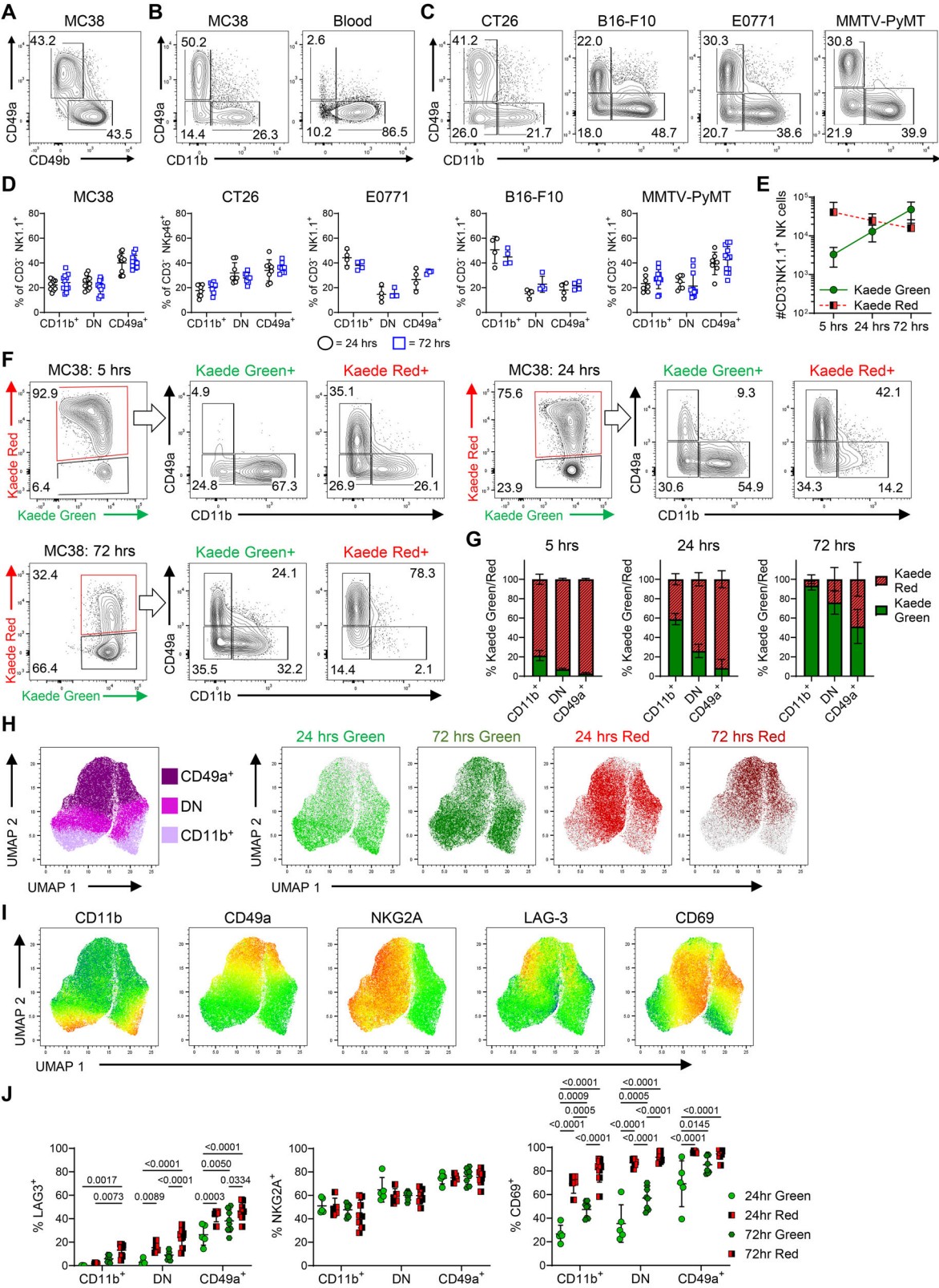

To validate some of the transcriptional changes observed, NK cells were isolated from photo-labeled tumors, stimulated ex vivo and assessed by intracellular flow cytometry. Analysis of NK cells at 24 h post-photoconversion revealed that production of CCL5 was restricted to KG+ CD11b+ CD49a- NK cells, confirming the rapid loss of CCL5 expression after tumor entry (Fig. 3A). NK cell production of IFNγ is a further mechanism by which innate immune cells may activate DCs to

enable priming of antigen specific CD8+ T cells[46]. Post-stimulation ex vivo, the intratumoral NK cells produced significantly less IFNγ than splenic NK cells (Fig. 3B). To further investigate NK cell *Ifng* expression in the absence of ex vivo stimulation, we turned to *Ifng*[cre/mKate2] reporter mice. To confirm accurate reporting in this newly generated model, splenocytes were stimulated ex vivo with recombinant IL-12 and IL-18, which resulted in robust mKate2 reporting of *Ifng* expression in NK

**Fig. 2 | Differential expression of CD11b and CD49a capture temporal changes in NK cell phenotype. A** Expression of CD49a vs CD49b by NK cells (CD3- NK1.1+ ) isolated from MC38 tumors grafted subcutaneously on the flank. **B** Expression of CD49a vs CD11b by NK cells isolated from MC38 tumors, alongside blood. **C** Expression of CD49a vs CD11b by NK cells across multiple tumor models. NK cells identified as CD3- NK1.1+ in C57BL/6 mice, CD3- NKp46+ in BALB/c mice. **D** Proportion of NK cells within the CD11b+ CD49a-, CD11b- CD49a- (double negative, DN) and CD11b- CD49a+ gates across the different tumor models. Data collected from B16-F10-OVA and EO771 tumors was from 1 independent repeat, and MC38, CT26, and MMTV-PyMT tumors pooled from 2 independent repeats, where MC38 (24 h $n = 12$, 72 h $n = 12$), CT26 (24 h $n = 8$, 72 h $n = 8$), EO771 (24 h $n = 4$, 72 h $n = 4$), B16-F10-OVA (24 h $n = 4$, 72 h $n = 4$), and MMTV-PyMT (24 h $n = 7$, 72 h $n = 12$). **E** Number of Kaede Green+ and Kaede Red+ NK cells at 5, 24 and 72 h post photoconversion of MC38 tumors. **F** Expression of CD49a versus CD11b by Kaede Green+ and Kaede Red+ NK cells at 5, 24 and 72 h post photoconversion. **G** The proportion of Kaede Green/Red for each NK cell subset at each time point post photoconversion. Data at 5 ($n = 5$) h is representative of 1 independent repeat, whereas 24 ($n = 8$) and 72 ($n = 11$) h pooled from 2 independent repeats. **H** UMAPs showing protein expression of CD11b and CD49a alongside Kaede Green/Red expression by NK cells isolated from MC38 tumors at 24 and 72 h post photoconversion. **I** UMAPS showing expression of CD11b, CD49a, NKG2A, LAG-3 and CD69. **J** Enumeration of the proportion of cells expressing NKG2A, LAG-3 and CD69 across the NK cell subsets. Data in UMAPs 24 h ($n = 5$), and 72 h ($n = 8$) are representative of two independent repeats. Statistical significance was determined by two-way ANOVA with Šidák's multiple comparisons test (**J**). Data are presented showing all individual data points as well as the mean value +/- SD. In all experiments, 'n' defines a single tumor on an individual mouse, i.e., $n = 3$ refers to 3 mice each with a single tumor.

cells but not T cells (Supplementary Fig. 8A). However, NK cells freshly isolated from MC38 tumors grafted into *Ifng*[cre/mKate2] mice, lacked mKate2 expression regardless of their integrin expression (Fig. 3C). To validate the changes in the cytotoxic profile of intratumoral NK cells indicated by the transcriptomic analysis (Fig. 1J), we confirmed granzyme A was produced by the vast majority of CD11b+ CD49a- cells, but significantly less was detected in the CD11b- CD49a- and CD11b- CD49a + populations (Fig. 3D). Contrasting with the decline in granzyme A production, and again consistent with the changes observed over transcriptional pseudotime, granzyme C production was lacking from KG+ CD11b+ CD49a- cells and most highly produced by KR+ CD11b- CD49a+ cells (Fig. 3E). granzyme B, which was not differentially expressed over pseudotime at the transcriptional level, was detected across all the NK cell populations and a significantly higher proportion of KR+ NK cells produced granzyme B (Fig. 3F). However, production of Perforin was significantly reduced in CD11b- CD49a+ NK cells versus the CD11b+ CD49a- populations (Fig. 3G), Finally, the proportion of NK cells expressing CD107a was significantly reduced amongst the KR+ compartment (Fig. 3H).

These changes in the function of NK cells when retained within tumors were further confirmed in other murine tumor models (Supplementary Fig. 8B–E). Of note, having analyzed CT26 tumors grown subcutaneously, we further asked whether dysfunctional NK cells were also observed when this colorectal cell line was grown orthotopically (Supplementary Fig. 8F, G). Here, in addition to EOMES+ NK cells within the CD3- NKp46+ populations, CXCR6+ EOMES- ILCs were evident, a population lacking in the subcutaneous model that reflected the contribution of ILCs present within the local tissue (Supplementary Fig. 8H, I). To directly assess whether cNK cells differentiate into the CD11b- CD49a+ phenotype, we MACS-isolated splenic NK cells from Kaede mice and transferred them i.v. into WT C57BL/6 hosts bearing MC38 tumors (Supplementary Fig. 9A, B). Only within the MC38 tumors did the transferred KG+ NK cells upregulate CD49a and granzyme C (Supplementary Fig. 9C–G). Finally, we sought to test whether the CD11b- CD49a+ NK population was less cytotoxic that the CD11b+ CD49a- cells that enter tumors. To do this, we FACS-isolated the CD11b+ and CD49a+ NK cells from MC38 tumors and tested their ability to lyse B16-F10 tumor cells in vitro over 4 h (Fig. 3I). The CD11b+ population showed significantly better killing than the labeled CD49a+ NK cells, consistent with functional differences in the cytotoxic ability of these populations (Fig. 3J). Collectively, these data demonstrate the presence of dysfunctional NK cells across multiple preclinical cancer models targeting different tissues. In all, upregulation of CD49a expression in the absence of CD11b is associated with retention in the TME, altered effector functions and less efficient tumor cell killing.

### Multiple mechanisms in the TME drive the conversion of cNK cells to a tumor - retained CD49a + NK cell compartment

Having defined the cellular state adopted by cNK cells that become retained within tumors, we sought to understand the mechanisms within the TME that drive this cell fate. The TGFβ and PGE₂ pathways have both been linked to altered NK cell functions both in vitro and in vivo[13,19,20]. We initially asked whether NK cells with the phenotype and functional repertoire of CD49a+ 'tumor-retained' cells, could be induced by exposure of cNKs to recombinant TGFβ and/or PGE₂ ex-vivo. Culture with TGFβ, but not PGE₂, caused the upregulation of CD49a, however CD11b expression did not change (Supplementary Fig. 10A). Culture with either TGFβ or PGE₂ alone, or in combination, diminished NK cell production of CCL5 upon restimulation, while complete loss of IFNγ expression required both TGFβ and PGE₂ exposure (Fig. 4A, Supplementary Fig. 10B). However, no alteration in granzyme A or granzyme C production was observed, indicating that the full spectrum of changes associated with the NK cells retained within tumors could not be recapitulated in vitro.

While we could partially recapitulate NK phenotype and functional changes in vitro, to further define the cues that modulate intratumoral NK cells we turned to in vivo approaches, reasoning that these would better model the complex TME. To investigate the role of TGFβ in driving changes to NK cells within the tumor, we sought to deplete intratumoral Tregs, given these are a critical source of TGFβ[47]. Since NK cells can express CTLA4 and CD25[48,49], we utilized anti-OX40 Abs, which efficiently deleted the FoxP3+ CD4 T cells within the tumor while leaving NK cells intact (Supplementary Fig. 10C–E). However, depletion of intratumoral Tregs did not alter the proportions of NK cells differentially expressing CD49a or CD11b (Supplementary Fig. 10F). No differences in IFNγ, granzyme A or granzyme C production were detected, although CCL5 production was significantly enhanced by Treg-depletion (Supplementary Fig. 10G).

To investigate the impact of tumor cell-derived PGE₂, MC38 cells deficient COX-1 and COX-2 (encoded by *Ptgs1*/Ptgs2, termed MC38[ptgs-/-]) were grafted on the flank of C57BL/6 Kaede mice and the NK cell compartment compared with that of MC38 tumors (Fig. 4B). Despite impaired tumor growth of the MC38[ptgs-/-] cells (Fig. 4C), the proportion of NK cells was not significantly altered, and the retention of NK cells only modestly impacted (Fig. 4D–F). While the proportion of the NK cell subsets defined by CD49a and CD11b expression were unchanged, significant differences in CCL5 and granzyme C production were observed, indicating a partial effect on NK cell phenotype and function (Fig. 4G, H).

Hypoxia is a common feature of solid tumors and has been linked with the altered NK cell effector functions[50,51]. To explore the potential effects of hypoxia on NK cells, we first repeated our in vitro cultures at 1% oxygen, however, only very modest changes in integrin and granzyme expression were observed (Supplementary Fig. 10H–L). To specifically test the role of HIF-1α in restraining NK cell functions within the tumor, we generated *Ncr1*[cre] x *Hif1a*[f/f] x Kaede mice and grafted these, alongside cre-negative littermate controls with MC38 tumors (Fig. 4I). Tumors were photoconverted on day 12 and analyzed 48 h later after ex vivo stimulation. Conditional deletion of HIF-1α expression in NK cells failed to significantly impair tumor growth (Fig. 4J) and

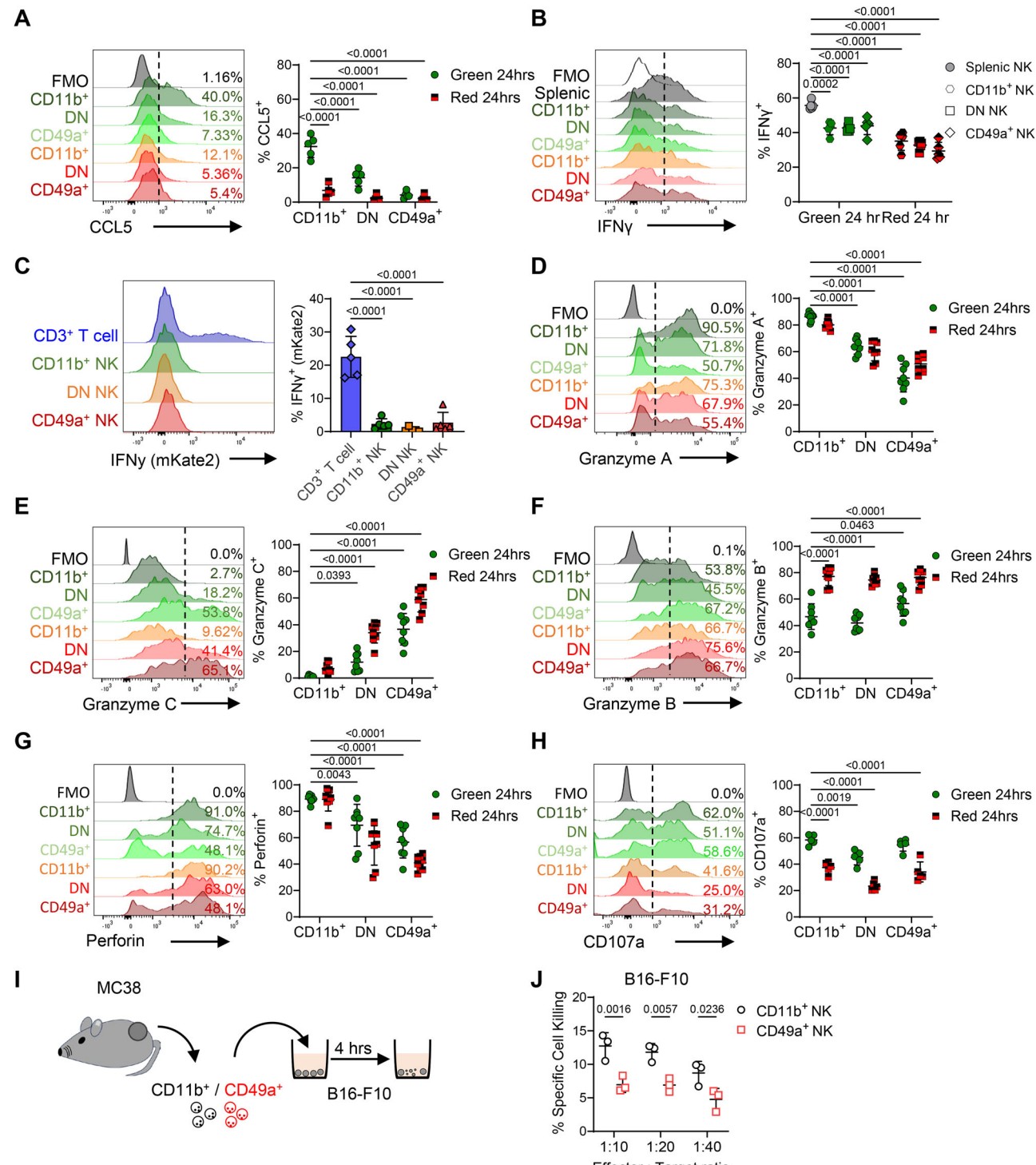

**Fig. 3 | NK cells rapidly change chemokine, cytokine and granzyme production within 24 h of entering the TME.** Flow cytometry was used to validate changes to the function of NK cells isolated from MC38 tumors at 24 h post photoconversion. **A** Proportion of NK cells producing CCL5 after ex vivo stimulation, with representative histograms alongside enumeration (*n* = 5). **B** Proportion of NK cells producing IFNγ after ex vivo stimulation, with representative histograms alongside enumeration (*n* = 5). **C** Reporting of mKate2 in using *Ifnycre/*^mKate2 reporter mice, with T cells versus NK cells isolated from MC38 tumors assessed. Representative histograms and the proportion of mKate2+ cells shown. Representative histograms showing proportion of NK cells producing (**D**) granzyme A, (**E**) granzyme C, (**F**) granzyme B, (**G**) CD107a, (**H**) Perforin in MC38 tumors that were photoconverted and analyzed 24 h later. Data pooled from 2 independent experiments (*n* = 8) for all

analyses except CD107a expression where *n* = 5 from 1 independent experiment. **I** Cartoon showing experimental setup whereby CD11b+ and CD49a+ NK cells from MC38 tumors were FACS-isolated and co-cultured with B16-F10 melanoma cells pre-treated with cell trace violet that were then stained for cell viability. **J** NK-mediated target cell killing comparing cytotoxicity between FACS sorted CD11b+ and CD49a+ NK cells (*n* = 3). Significance was determined by two-way ANOVA with Šidák's multiple comparison test (**A**, **B**, and **D**–**H**) comparing means to Kaede Green+ CD11b+ cells, or between groups (**J**), and a Kruskal–Wallis test with Dunn's multiple comparison test (**C**). Data are presented showing all individual data points as well as the mean value +/- SD. In all experiments 'n' defines a single tumor on an individual mouse, i.e., *n* = 3 refers to 3 mice each with a single tumor.

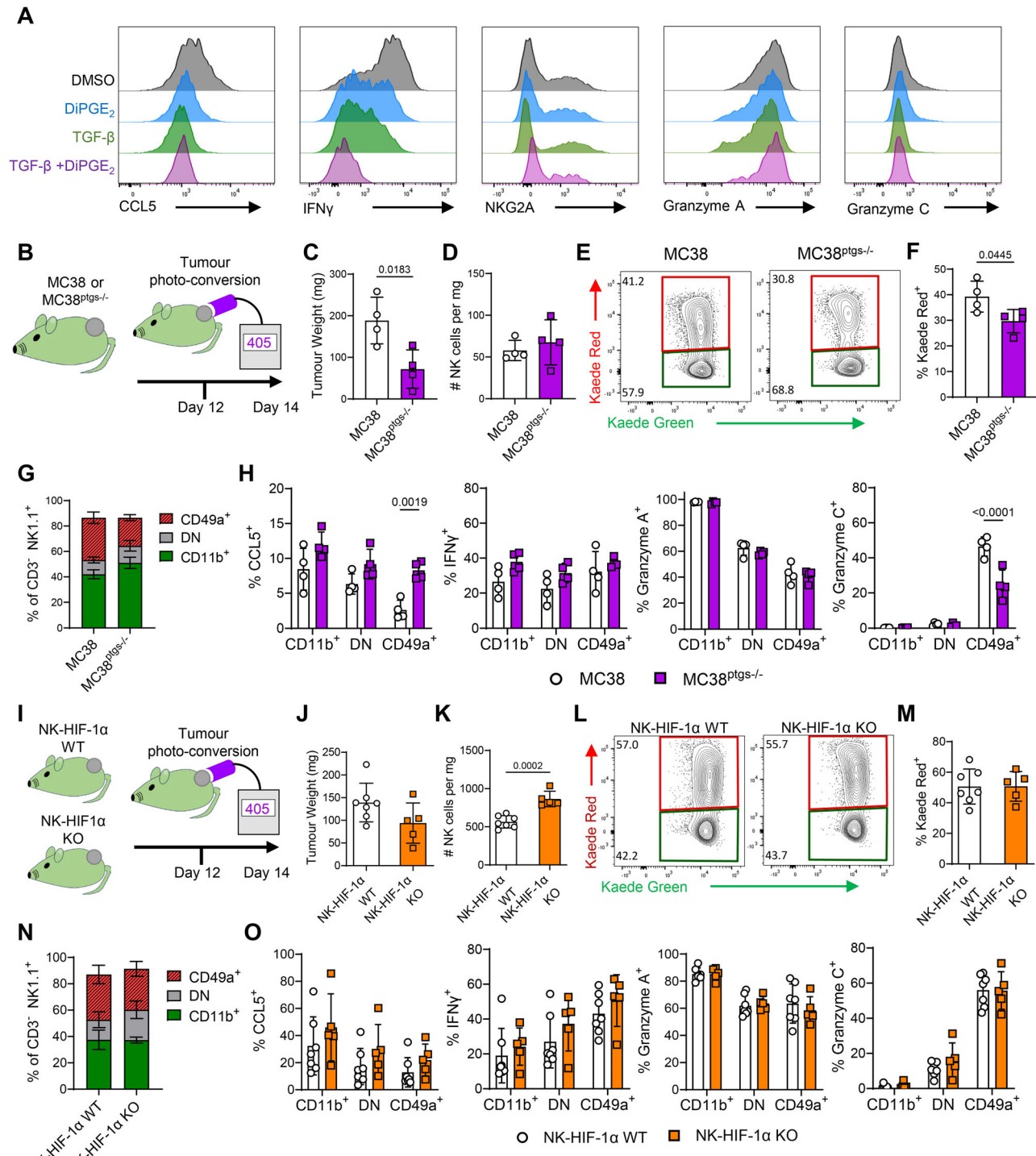

**Fig. 4 | Multiple mechanisms in the TME drive the conversion of cNK cells to a tumor-retained CD49a+ state.** The role of TGFβ, PGE₂ and HIF-1α were investigated in vitro and in vivo as mechanisms promoting NK cell differentiation to the tumor retained state characterized by CD49a expression and disrupted core functions. **A** Bar chart showing proportion of CD11b+ and CD49a+ NK cells after 48 h in culture in RPMI with IL-2/IL-15 further supplemented with TGF-β and/or DiPGE₂. **B** Cartoon summarizing experimental design for impeding tumor produced PGE₂. Kaede mice were grafted with either MC38 (*n* = 4) or MC38^ptgs1-/- ptgs2-/- (MC38^ptgs-/-, *n* = 4) cells, photoconverted on D12 and analyzed 48 h later. **C** Tumor weight. **D** Enumeration of NK cells per mg tumor. **E** Flow cytometry plots showing Kaede Green versus Kaede Red expression by NK cells. **F** Proportion of NK cells expressing Kaede Red label 48 h post photoconversion. **G** Proportion of NK cells in CD11b+ CD49a-, CD11b- CD49a- and CD11b- CD49a+ subsets. **H** Proportion of NK

cells producing CCL5, IFNγ, granzyme A, granzyme C after ex vivo restimulation. **I** Cartoon summarizing experimental design for targeting *Hif1a* within NK cells, using *Ncr1*^cre x *Hif1a*^f/f x Kaede mice (NK-HIF1α KO *n* = 5) versus cre-negative littermates (NK-HIF1α WT *n* = 7) grafted with MC38 tumors, photoconverted on D12 and analyzed 48 h later. **J** Tumor weight. **K** Enumeration of NK cells per mg tumor. **L** Flow cytometry plots showing Kaede Green versus Kaede Red expression by NK cells. **M** Proportion of NK cells expressing Kaede Red label 48 h post photoconversion. **N** Proportion of NK cells in CD11b+ CD49a-, CD11b- CD49a- and CD11b-CD49a+ subsets. **O** Proportion of NK cells producing CCL5, IFNγ, granzyme A, granzyme C after ex vivo restimulation. Statistical significance was determined by unpaired *t* tests (**C**, **D**, **F**, **J**, **K**, **M**) or two-way ANOVA with Šidák's multiple comparison test comparing group means (**H**, **O**). Data are presented showing all individual data points as well as the mean value +/- SD.

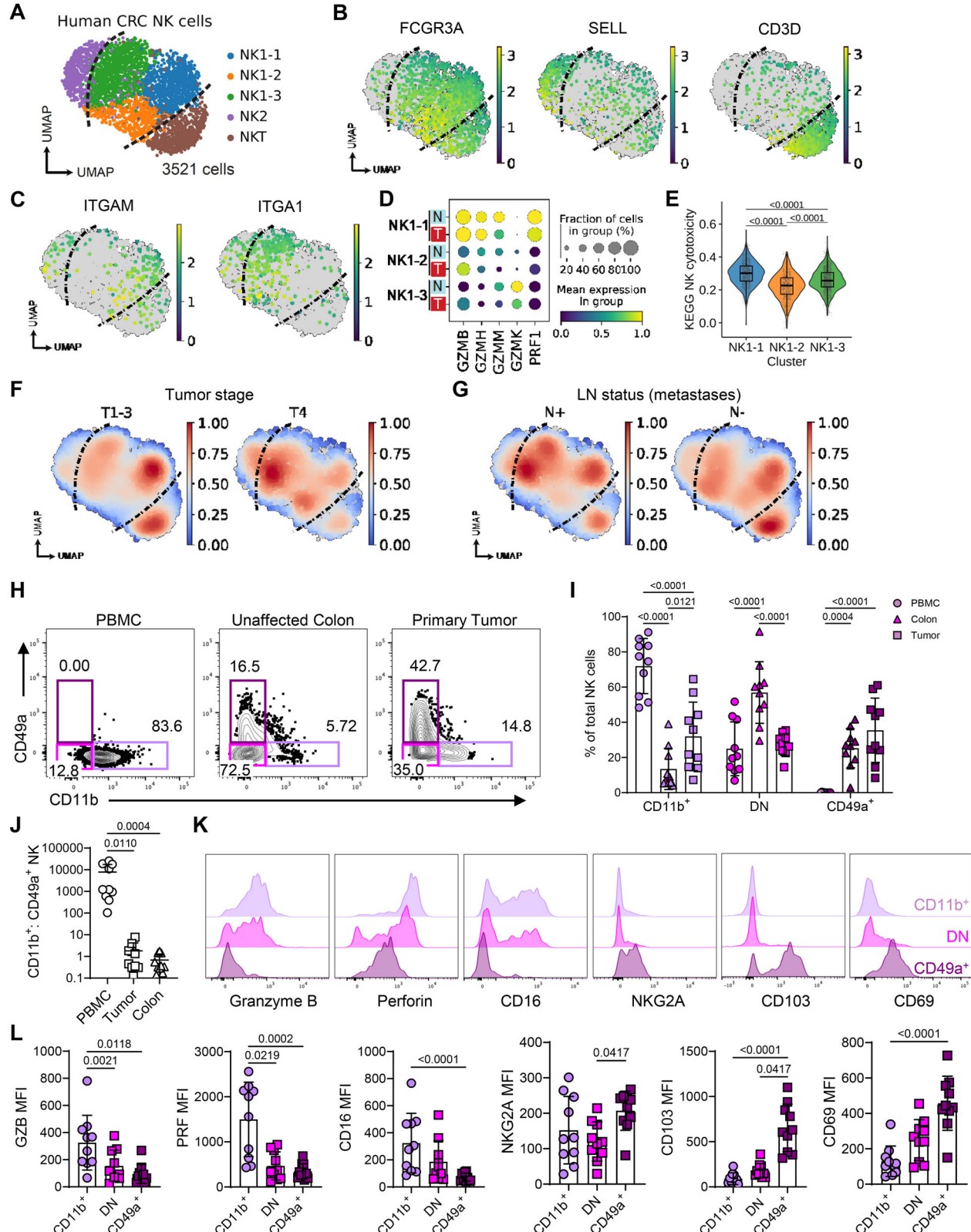

while there was a modest increase in the number of NK cells in the tumor (Fig. 4K), no differences in the % of KR+ NK cells, the proportion of NK cell subsets, nor production of CCL5, IFNγ, granzyme A or granzyme C was detected (Fig. 4L–O).

Collectively, these data recapitulate published observations regarding the role of TGFβ and PGE₂ in altering NK cell function, particularly CCL5 production. However, numerous signals, rather than a single mechanism appear to cause the full array of changes characteristic of NK cells that become retained within murine tumor models.

## NK cells in human colorectal cancer show loss of effector functions

Having demonstrated across multiple murine cancer models that the NK cells retained within tumors become dysfunctional, we sought

**Fig. 5 | NK cells in human CRC show loss of effector functions.** Evidence for dysfunctional NK cells within human CRC was sought through bioinformatics analysis of publicly available data sets alongside flowcytometric analysis of primary human CRC samples. **A** UMAP of 3521 NK cells from scRNA-seq of 62 human CRC samples, from GSE178341. **B** Expression of selected marker genes for clusters shown in 'A'. **C** Expression of *ITGAM* and *ITGA1* in NK1 subsets. **D** Granzyme and perforin gene expression in NK1 subsets, in tumor (T) versus normal adjacent (N) tissue. **E** Gene set enrichment for KEGG Natural killer cell-mediated cytotoxicity between NK1 clusters. **F** Gaussian kernel density embedding of cells in CRC tumors by tumor staging, and (**G**) lymph node (LN) metastases. **H** Representation flow plots depicting the frequency of CD49a+ and CD11b+ NK cells across PBMC, unaffected colon tissue and CRC primary tumor for the same donor. **I** Bar plot showing the frequency of CD11b+ CD49a-, CD11b- CD49a- and CD11b- CD49a+ NK cells across compartments. **J** Ratio of CD11b+ CD49a- to CD11b- CD49a+ NK cells across tissue compartments. **K** Representative histograms showing expression of selected markers on tumor infiltrating NK cell subsets. **L** Bar plots depicting geoMFI values for specific markers in tumor infiltrating NK cells. Statistical significance was determined by: (**E**) two-sided Wilcoxon rank-sum test, data are shown as box (median; box, 25th percentile and 75th percentile; whiskers, 1.5*inter-quartile range) and violin plots, (**I**) two-way ANOVA with Tukey's multiple comparison test, and (**J, L**) Friedman test with Dunn's multiple comparisons test. Data are presented showing all individual data points as well as the mean value +/- SD. **H–L** describe data from 10 patients with colorectal cancer, where *n* = 10 and 'n' defines an individual patient.

evidence that NK cells undergo a comparable transition in human tumors. Deconvolution of immune cells from bulk transcriptomes of 521 human colorectal cancers indicated that NK cells are present in colorectal tumors, but most adopt a "resting", non-activated state (Supplementary Fig. 11A, B)[52]. To precisely characterize their phenotype, we re-analyzed scRNA-seq of human colorectal tumors[53], including 3521 NK cells from 62 patients (Fig. 5A, Supplementary Fig. 11C, D). Among these, the majority were *FCGR3A*-expressing, *SELL*-negative, *CD3D*-negative cells (termed "NK1"), which are the CD56dimCD16+ cytotoxic NK cell subset in humans[54] (Fig. 5B). Within NK1 cells, unbiased clustering revealed 3 distinct cell states, including an *ITGAM*-expressing NK1-1 subset, an *ITGA1*-expressing NK1-3 subset, and a double-negative NK1-2 subset, similar to our observations in murine tumors (Fig. 5C). Expression of several granzymes (*GZMB, GZMH, GZMM*), perforin (*PRF1*), and gene signatures representing NK cytotoxicity and activation were enriched on NK1-1 but downregulated in NK1-2 and NK1-3 (Fig. 5D, E, Supplementary Fig. 11E). This is consistent with our observations in mice, where resident CD49a+ cells rapidly downregulate cytotoxicity and activation markers compared to newly infiltrating CD11b+ cells. Moreover, there was reduced expression of *PRF1* and NK cytotoxicity genes in tumor-infiltrating NK cells versus adjacent normal colorectal tissue, further suggesting a loss of anti-tumor NK cell function (Fig. 5D, Supplementary Fig. 11F). The *ITGAM*-expressing subset more closely associated with early-stage T1-3 tumors, while the *ITGA1*-expressing were enriched within advanced-stage T4 tumors or tumors that had progressed to disseminate lymph node metastases (Fig. 5F, G). Finally, we compared the DEGs between *Itgam*-expressing and *Itga1*-expressing tumor NK cells in this human CRC data with our original murine NK cells from MC38 tumors. This analysis identified 184 DEGs, which included *Ccl5, Ifng Il2rb, Tnfrsf1b* and *Eomes* as genes upregulated in the CD11b+ NK cells and *Cd96, Cd160, Havcr2, Klrg1, Tbx21, Gzmc* as genes upregulated in the CD49a+ NK cells (Supplementary Fig. 11G, Supplementary Data 1).

Thus, a spectrum of NK cell transcriptional states, comparable to that observed in murine tumor models, was evident in human CRC. These data are consistent with recent transcriptomic analyses across multiple human tumors, which identified tumor-associated NK cells with impaired effector functions and that associated with unfavorable prognosis[55]. To validate these phenotypes at the protein level, we compared NK cells in peripheral blood, with those obtained from unaffected colon and primary CRC. Analysis of paired tissue samples from 10 patients with CRC demonstrated a clear shift from predominantly CD11b+ NK cells in blood to more CD49a+ NK cells in the tumors, while normal colon was dominated by NK cells expressing neither CD11b nor CD49a (Fig. 5H–J, Supplementary Fig. 12). Deeper characterization of the phenotype of NK cell subsets within the tumor showed a lower expression of the cytotoxicity-mediating molecules granzyme B and perforin as well as CD16 within the CD49a+ subset (Fig. 5K, L). This was paralleled by higher expression of NKG2A and the tissue-residency markers CD103 and CD69 on CD49a+ NK cells.

Collectively, these data demonstrate the presence of a CD49a+ subset of NK cells with features of tissue residency that lacks the expression of cytotoxic proteins within the CRC TME.

## Enhanced IL-15 signaling drives formation of a distinct cytotoxic intratumoral NK cell population

Our analyses above demonstrate that cNK cells recruited into the tumor rapidly become dysfunctional, both in their cytotoxicity and their capacity to recruit and activate DC in the tumor. While depletion of NK cells prior to establishing lung metastases has been reported to result in a significant increase in disease burden[10,56], we questioned whether the dysfunctional NK cells that form in established tumors continued to contribute to the control of tumor growth. To test this, MC38 tumors were established in WT mice and then NK cells were depleted using anti-NK1.1 Abs, administered from 6 days of tumor growth. Highly efficient NK cell depletion was achieved within the tumor; however, the loss of intratumoral NK cells in established tumors did not alter growth across different tumor models (Supplementary Fig. 13A–G), even in the absence of T cells (Supplementary Fig. 13H–K). These data are consistent with the conclusion that intratumoral NK cells fail to meaningfully contribute to the anti-tumor response due to the rapidly acquired dysfunctional state.

Finally, we sought approaches that could block or reverse the loss of NK cell function imprinted by residence in the TME and thus support enhanced anti-tumor responses. IL-15 drives the differentiation and activation of both T and NK cells[57]. Initial patient data suggest that the response to IL-15R agonizts is associated with NK cell and CD8+ T cell expansion and increased IFNγ production. However, it is unclear how IL-15R agonism enhances NK cell anti-tumor responses or the potential recruitment of NK cells into solid tumors. To determine whether enhanced IL-15 signaling resulted in the maintenance of NK functions normally disrupted by adaptation to the TME, tumor bearing mice were treated with complexes of IL-15:IL-15Ra, since these complexes have a significantly enhanced activity versus recombinant IL-15 alone. We tested these complexes, administered at D7 and D12 of tumor growth, both prior to and concurrent with photoconversion, in multiple tumor models (Fig. 6A). BALB/c Kaede mice bearing CT26 tumors showed significantly enhanced tumor control with treatment (Fig. 6B). Analysis of the intratumoral NK cell compartment revealed striking differences in integrin expression following administration of IL-15:IL-15Ra complexes (Fig. 6C). Surprisingly, treatment resulted in the majority of the NK cells expressing CD49a, however we also observed the emergence of a CD49a+ CD11b+ population barely detectable in the PBS controls (Fig. 6D). To ask whether IL-15:IL-15Ra complexes caused the emergence of CD49a+ CD11b+ cells after entry into the tumor, or if these cells were formed in the periphery and trafficked into the tumor, we photoconverted tumors and compared the % KR expression within different intratumoral NK cell subsets. Treatment did not cause a significant difference in the total proportion of KG+ NK cells in the tumor (Fig. 6E). Analysis of the NK cell subsets defined by

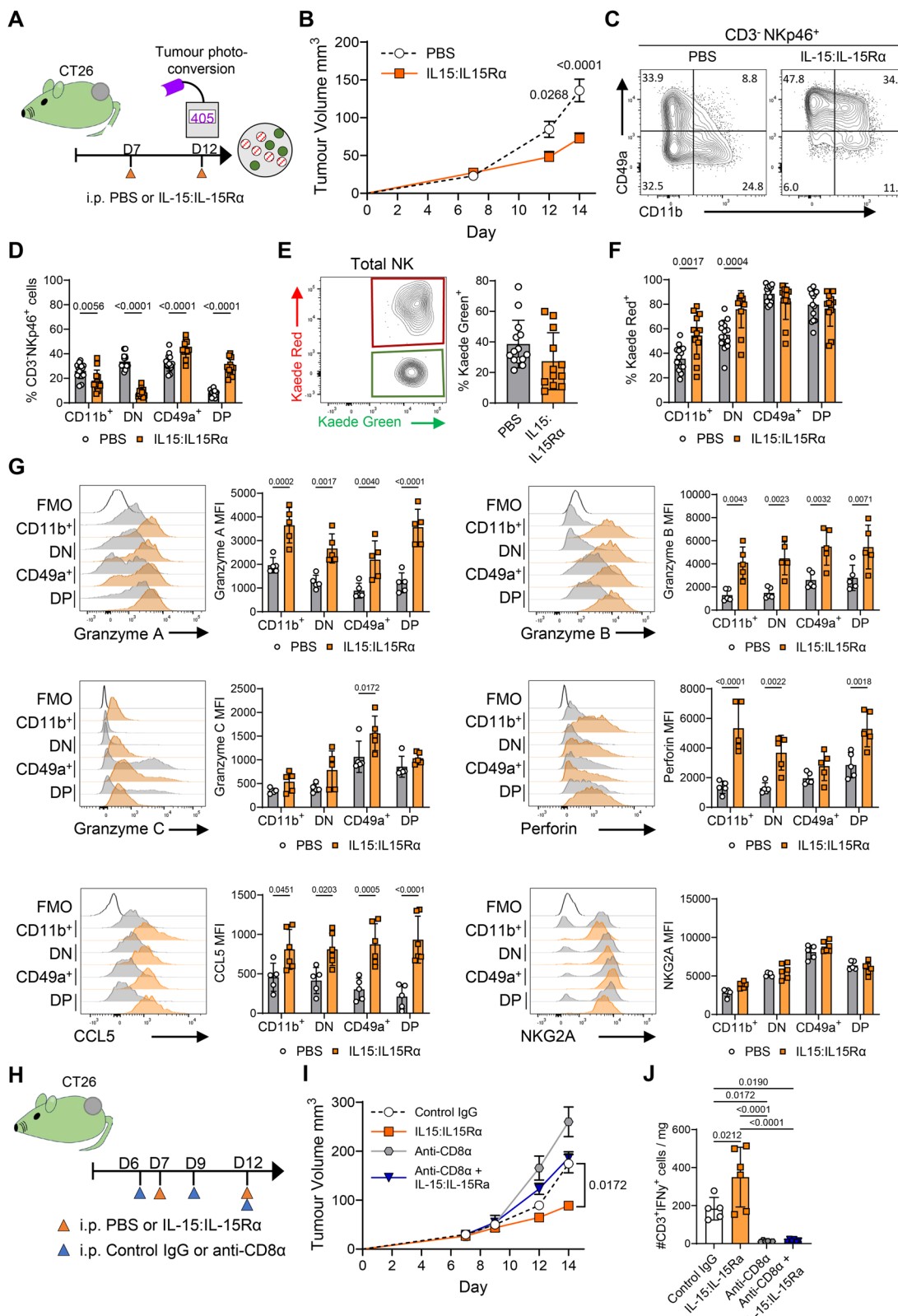

CD49a versus CD11b expression revealed that the vast majority of CD49a+ CD11b+ NK cells were KR+, demonstrating that this subset emerged over time in the tumor, rather than arising from cells trafficking into the tumor (Fig. 6F). Analysis of NK cell functions revealed that the expanded CD49a+ CD11b- and CD49a+ CD11b+ populations observed after IL-15:IL-15Ra treatment had robust expression of perforin and granzymes A, B, and C (Fig. 6G). Enhanced IL-15 signaling also

significantly increased CCL5 production, in the absence of dramatic changes in NKG2A expression (Fig. 6G). Comparable data was observed in both MC38 and B16-F10-OVA tumors after IL-15:IL-15Ra treatment (Supplementary Fig. 14). Finally, since enhanced IL-15 signaling impacts both CD8 T cells and NK cells, we sought to determine the NK cell contribution to the enhanced control of tumor growth after treatment. BALB/c Kaede mice bearing CT26 tumors were split into

**Fig. 6 | Administration of IL15:IL15Rα complexes result in enhanced tumor control and the formation of CD49+ CD11b+ intratumoral NK cells with heightened functions.** To block loss of NK cell functions within the tumor, BALB/c Kaede mice bearing CT26 tumors were treated with IL-15:IL-15Ra complexes. **A** Cartoon showing experimental design. **B** Growth curves for CT26. **C** Representative flow cytometry plots expression of CD49a versus CD11b by NK cells (CD3- NKp46+ ). **D** Proportion of NK cells with CD11b+ CD49a-, CD11b- CD49a-, CD11b- CD49a+ , CD11b+ CD49a+ phenotype. **E** Representative flow cytometry plot showing Kaede Green versus Kaede Red expression by total NK cells alongside enumeration of % Kaede Green+ NK cells. **F** Proportion of Kaede Red+ cells within CD11b+ CD49a-, CD11b- CD49a-, CD11b- CD49a+ , CD11b+ CD49a+ subsets of NK cells. Data pooled from 3 independent experiments, (cumulative totals: PBS $n = 14$, IL15:IL-15Ra $n = 12$). **G** Representative histograms and enumeration of granzyme A,

granzyme B, granzyme C, perforin, CCL5, and NKG2A (MFI) after ex vivo stimulation (data from 1 of 3 experiments shown, PBS $n = 5$ and IL-15:IL-15Rα $n = 5$). **H** Cartoon showing experimental design for CD8 T cell depletion in combination with IL-15:IL-15Ra complexes (PBS + IgG $n = 5$, PBS + anti-CD8α $n = 6$, IL-15:IL-15Rα + IgG $n = 6$, 15:IL-15Rα + anti-CD8α $n = 6$). **I** Tumor growth curve for experiment illustrated in (**H**). **J** Total number of NK cells per mg tumor. Statistical significance was determined by two-way ANOVA with Šidák's multiple comparisons test (**B**, **D**, **F**, **G** and **I**), unpaired $t$ test (**E**), or one-way ANOVA with Tukey's multiple comparisons test (**J**). Data are presented showing all individual data points as well as the mean value +/- SD, except (**B**, **I**) where mean value +/- SEM are shown. In all experiments 'n' defines a single tumor on an individual mouse, i.e., $n = 3$ refers to 3 mice each with a single tumor.

PBS or IL15:IL-15Ra treatment groups, and then further divided into groups receiving isotype control or anti-CD8α Abs to specifically deplete the CD8 T cell compartment (Fig. 6H). Administration of IL-15:IL-15Ra complexes combined with isotype control Abs mediated the greatest control of tumor growth, while CD8 T cell depletion resulted in faster tumor growth than in PBS control mice given isotype Abs (Fig. 6I). Depletion of CD8 T cells in combination with IL-15:IL-15Ra complexes improved tumor control versus PBS/anti-CD8α Abs, consistent with a distinct NK cell and CD8 T cell contribution to anti-tumor responses. Effective depletion of CD8 T cells was confirmed by analysis of IFNγ producing T cells within the tumor (Fig. 6J).

Combined these data reveal that enhanced IL-15 signaling is able to block the loss of core NK cell functions that occurs as cNK cells adapt to the TME. Administration of IL-15:IL-15Ra complexes results in the emergence of a distinct CD49a+ CD11b+ NK cell compartment, not normally present within tumors, that retains a highly cytotoxic profile and contribute to improved tumor control.

## Discussion

Here, we exploit the site-specific temporal labeling afforded by Kaede photoconvertible mice to determine how NK cells change over time spent within the TME. Multiple studies have described the presence of a distinct CD49a+ NK cell population within murine and human tumors[19–21,58,59]. Our data establishes that this population can be rapidly formed from cNK cells recruited from the circulation that then respond to conditions within the tumor to develop a tumor-retained state, distinct from that of circulating cNK cells or ILC1s within lymphoid or non-lymphoid tissues. Adaptation to the TME bears hallmarks of tissue-residency, including CD49a and CD69 expression, alongside the loss of the basic cNK effector functions of efficient cancer-cell killing and the recruitment and activation of DCs. Having defined the end-state of NK cell differentiation within tumors, we sought to determine the mechanisms that drive these changes. Manipulation of TGFβ, PGE₂ or the transcription program controlled by HIF-1α, was insufficient to drive the full array of changes that define tumor-retained NK cells, implying that the coordinated action of multiple signals orchestrates this fate. The loss of NK cell functions within the tumor could be prevented through enhanced IL-15 signaling, which pushed intratumoral NK cell differentiation towards a hybrid state defined by high CD49a expression, but also enhanced effector functions. Collectively, these data provide insight into how NK cells change upon recruitment into tumors, the rate at which this occurs, and their likely functions within the TME.

Particularly striking within our data, was the speed at which cNK cells entering the tumor lost core effector functions. While cNK cells appear to be continuously recruited into the tumor, 24 h within the growing tumor was sufficient for extensive transcriptomic reprogramming. By establishing the rate at which NK cells change, our data indicates that increasing the contribution of innate effector cells within the tumors may depend on over-riding mechanisms operating within the TME, rather than simply enhancing recruitment. Our data

further suggests that cellular therapies such as CAR NK cells may be less efficient than hoped in solid cancers and rapidly 'turned-off', if efforts to over-ride tissue-adaptation mechanisms are not appropriately considered. NK cell chemokine and cytokine production appear particularly sensitive to signals within the TME. CCL5 production was lost in vitro in the presence of either recombinant TGFβ or PGE₂, and could be enhanced in vivo upon targeting either intratumoral Tregs or tumor cell derived PGE₂. While cNK cells are potent producers of IFNγ, newly developed *Ifng* reporter mice indicate that intratumoral NK cells, regardless of their integrin expression, express very little of this key effector cytokine.

A complete mechanistic understanding of how the tumor-retained state is established within the tumor eluded us. Local signaling by TGFβ is well established in driving tissue-residency in both NK cells and CD8 T cells[19,60–62]. However, our data indicate that TGFβ signaling alone is insufficient to account for all the changes observed within the retained NK cell compartment, including the clear switch in granzyme production. Surprisingly, the predicted changes in transcription factor regulons over time failed to identify any upregulated pathways that might drive the transition to the CD49a+ state. Expression of *Hif1a* over pseudotime was evident, however, its targeted deletion in NK cells failed to change NK cell fate within the tumor models we assessed. Currently, our data points to conditions within the TME resulting in a dampening of many pathways that drive NK cell activation and expansion. Precisely how this is orchestrated remains an outstanding question.

Whether the CD49a+ NK compartment actively contributes to the anti-tumor response, or perhaps even impedes it, is worthy of further investigation, particularly since our data argues that this is the fate of all NK cells retained within the tumor. While we have concentrated on the loss of effector functions, in silico prediction of cellular interactions suggest that the CD49a+ NK cells may support other innate populations that inhibit the response. Detailed analysis of where the CD49a+ NK cells reside within the tumor and whether specific niches foster these cells could provide further insight into how these cells are sustained and their potential roles. Future studies of how changes in NK cell function are regulated should also consider whether the conversion of cNK to a tumor-retained state is reversible and the extent to which epigenetic changes underpin this transition.

We initiated these studies of the intratumoral NK cell compartment to better understand how this population was formed and the potential contribution of tissue-resident ILCs. Our data demonstrate that in multiple commonly used tumor models, the CD49a+ NK compartment is derived almost exclusively from cNK cells recruited from the circulation. Although the time frame afforded by photo-labeling is too short to fully assess tissue-residency, CD49a+ intratumoral NK cells also upregulated CD69, consistent with becoming resident within the tumor. Multiple studies in both mouse and human have previously linked CD49a expression with tissue-residency[63–65]. Importantly, in our data, this population remains transcriptionally distinct from the limited bona fide ILC population we could detect. Our interpretation is

that cNK cells establish a 'tumor-retained' state that can be partly defined by CD49a expression, rather than 'converting' into ILC1s[19]. At the cellular level, we struggled to identify a clear CD127+ ILC population by flow cytometry in any of the syngeneic cell line tumor models, limiting further investigation. The lack of CD127+ ILCs within these syngeneic tumors likely reflects the propagation of these tumors within a 'space' under the skin, rather than a specific tissue with an associated ILC compartment. Recent studies used fate-mapping of *S1pr5* expression to conclude that most intratumoral CD49a+ innate lymphocytes were not derived from cNK cells[33]. Here, a spontaneous MMTV-PyMT breast tumor model[66] was utilized and the presence of ILCs within the mammary fat pad, may explain the differences observed compared with our tumor models. Of note, while ILCs were largely absent from CT26 tumors grown subcutaneously, a CXCR6+ EOMES- ILC population that expressed granzyme C was evident within orthotopic CT26 tumors, again suggesting that the tissue site matters. Further studies using fate-mapping of *S1pr5* and *Gzmc*, alongside other approaches to distinguish NK and ILC1 such as *Rora* reporting[36], should be employed across the multiple murine cancer models to further resolve contribution of differentiating cNKs cells versus tissue-residents ILCs.

Having defined the immunological cost of NK cells adapting to the TME, we sought to identify interventions that could limit or block the loss of key functions. IL-15 promotes NK cell survival, proliferation, and cytotoxicity, and therapeutically enhanced IL-15 signaling, alone or in combination with other targets such as PD-L1, is currently being trialed[67–70]. Provision of IL-15 within the TME likely involves CCR7+ DCs, which express high levels of IL-15 and IL-15RA, co-localize with NK cells in human cancers and can support further molecular interactions such as PVR:TIGIT[55,71,72]. Therapies utilizing IL-15R agonizts (ALT-803 and NIZ985) are currently being trialed alone or in combination with ICB (NCT02523469, NCT02452268, NCT03228667, NCT03520686, NCT05096663) and have shown some initial promise in patients that have previously relapsed from immune checkpoint inhibitor treatment[67]. While recombinant IL-15 failed to impact tumor growth in our hands, IL-15:IL-15Ra complexes which drive superior in vivo signaling[73], were able to consistently curtail tumor growth in multiple cancer models. Common to all the treated tumors was the upregulation of CD49a expression on intratumoral NK cells, enhanced CCL5 production, and increased granzyme and perforin expression consistent with heightened cytotoxicity and their contribution to the enhanced anti-tumor response. Depletion of CD8 T cells within these tumors indicated a contribution from the revived NK compartment alongside a heightened CD8 T cell response. While systemic administration of IL-15:IL-15Ra complexes did drive splenomegaly, due to peripheral expansion of peripheral NK and CD8 T cell populations[73,74], our data indicates that treatment promoted both a tissue-resident phenotype and superior effector function for NK cells specifically within the tumor. Previous studies have identified that the combination of IL-15 and TGFβ signaling synergized in driving a tissue-resident phenotype defined by CD49a, CD69 and CD103[75]. Our data highlights that the strong activation driven by IL-15:IL-15Ra complexes sustains heightened effector functions despite seemingly promoting aspects of a tissue-residency program.

In summary, here we have provided a detailed map of how circulating NK cells respond to the TME and their fate within this tissue. These studies highlight how quickly immune cells can become dysregulated within the TME and can support the design of therapeutic approaches designed to revive the innate arm of the anti-tumor response and enhance tumor control.

## Methods
### Study design
The main aims of this study were to understand how the NK cell compartment of tumors was formed, determine the mechanisms

controlling the fate of NK cells within the tumor and to identify approaches to manipulate NK fate and function within the tumor. We sought to define the fate of NK cells entering the tumor from the circulation through site-specific temporal labeling of the entire immune compartment of tumors. This was achieved using syngeneic tumor cell lines grafted into transgenic mice expressing green photoconvertible proteins which, when exposed to violet light, switched to a red fluorescence that could then be detected for a number of days post photoconversion. Tumors were colonized by the host immune cells (which were 'green') and by labeling all the immune cells within the tumor at a given moment in time (i.e., turning them 'red'), we could distinguish retained and newly entering cells to then unpick the fate of cells over time in the tumor. We use time course experiments and a combination of single-cell RNA-sequencing, to unbiasedly capture transcriptomic changes, followed by flow cytometry to validate these changes at the protein level. Building from this characterization, we then investigated the mechanisms causing the changes in NK cell phenotype and function as well as interventions that could alter the changes that occurred in NK cells after entering the tumor.

### Mice
Female C57BL/6J mice, aged 6–8 wks were purchased from Charles River (Strain code 632) and allowed to acclimatize to the University of Birmingham Biomedical Services Unit for a week prior to use in experiments. Rag2[-/-] mice (JAX, strain code 008449) were bred at the University of Birmingham Biomedical Services Unit. C57BL/6 Kaede[35] and BALB/c Kaede mice are maintained and bred at the University of Birmingham Biomedical Services Unit. *Hif1a*[fl/fl]*Ncr1*[iCreTg] mice[51] (kindly provided by Dr. Christian Stockmann, University of Zurich, Switzerland) were crossed with C57BL/6 Kaede mice and maintained at the University of Birmingham Biomedical Services Unit. Ifng[cre/mKate2] mice were generated by Taconic and maintained at the University of Birmingham Biomedical Services Unit. For all mice generated and maintained at the University of Birmingham Biomedical Services Unit, both male and female mice were used in experiments, although within an experiment, mice were sex-matched. Across the project, experimental mice were aged between 8 and 18 weeks. Animals were maintained under specific pathogen-free conditions and used in accordance with Home Office Guidelines under a Project License awarded to D.R. Withers and approved by the University of Birmingham Animal Welfare and Ethical Review Body. Control animals were co-housed. Mice were killed by cervical dislocation at the end of the experiment.

### Generation of Ifng[cre/mKate2] reporter mice
C57BL/6NTac-*Ifng*[em6630(iCre-mKate2)Tac] mice were generated by Taconic, An improved Cre (iCre)[76] and the far-red fluorescent reporter mKate2[77] were inserted into the *Ifng* locus immediately after exon 4, flanked by T2A and P2A self-cleaving peptides.

### Mouse tumor models
MC38 (kindly provided by Dr. Gregory Sonnenberg, Weill Cornell Medicine, New York, NY), MC38-Ova (obtained from AstraZeneca), CT26 (kindly provided by Professor Tim Elliot, University of Oxford, Oxford, UK), MC38[ptgs-/-] (kindly provided by Dr Santiago Zelenay, University of Manchester, Manchester, UK) murine colon adenocarcinoma cells, and B16F10-Ova (obtained from AstraZeneca) murine melanoma cells were cultured in RPMI supplemented with 2 mM L-glutamine (#21875034 Thermo Fisher Scientific), 10% FBS (#F9665 Sigma-Aldrich), and penicillin-streptomycin (#P4333 Sigma-Aldrich). EO771 (kindly provided by Dr. Fedor Berditchevski, University of Birmingham, Birmingham, UK), and 4T1 (obtained from AstraZeneca) mammary carcinoma cells were cultured in DMEM supplemented with 2 mM L-glutamine (#21875034 Thermo Fisher Scientific), 10% FBS (#F9665 Sigma-Aldrich), and penicillin-streptomycin (#P4333 Sigma-Aldrich). All cells were cultured at 37 °C with 5% $CO_2$ before being

harvested and suspended in Dulbeccos' PBS (#D8662 Sigma-Aldrich) for tumor injection. $1 \times 10^4$ 4T1, or $1 \times 10^5$ EO771 tumor cells were injected in 50 µl into the mammary fat pad of female mice under anesthesia via 2% gaseous isoflurane. $2.5 \times 10^5$ (CT26, MC38, MC38-Ova, MC38$^{ptgs-/-}$), or $5 \times 10^5$ B16-F10-Ova tumor cells were injected in 100 µl subcutaneously into the pre-shaven left flank under anesthesia via 2% gaseous isoflurane. For orthotopic CT26 injections, $1 \times 10^7$ CT26 tumor cells were injected in 25 µl PBS into the colonic wall using an endoscope fitted with an integrated multi-purpose rigid telescope (Karl Storz Endoskope) under anesthesia via 2% gaseous isoflurane. Tumor size was periodically measured with a digital Vernier caliper, and the volume was calculated using the formula $V = L \times (W)^2 \times 0.52$ in cubic millimeters, where L represents the longest diameter and W the perpendicular diameter to L for the tumor. Tumor weights were measured at the endpoint of the experiment. Mice were sacrificed 5 h, 24 h, 48 h, or 72 h post-photoconversion and tumors were harvested for analysis.

## Cell depletion

Depletion of NK cells and CD8+ T cells was achieved by administering anti-NK1.1 (PK136, 200 µg) or anti-CD8α (53–6.7, 400 µg), or InVivoMab mouse IgG2a isotype control (BioXCell, C1.18.4, 200 µg), or InVivoMab rat IgG2a isotype control (BioXCell, 2A3, 400 µg) in PBS via intraperitoneal injection every 3 days beginning on day 6 post tumor engraftment. Depletion of Tregs was achieved by administering anti-OX40 (OX86, 200 µg), or PBS via intraperitoneal injection on day 7 and 11 of tumor growth. Anti-NK1.1, anti-CD8α, and anti-OX40 antibodies were provided by AstraZeneca.

## Administration of IL-15:IL-15Rα complexes

rIL-15 (#210-15, Peprotech) and rIL-15Rα (#551-MR-100) were reconstituted in PBS at a ratio of 1:5 and incubated at 37 °C for 30 min to form IL-15:IL-15Rα complexes. IL-15 complexes (2.5 µg rIL-15:12.5 µg rIL-15Rα) were administered via intraperitoneal injections on day 7 and 12 of tumor growth.

## Splenic NK cell transfer

NK cells were enriched from the spleens of C57BL/6 Kaede mice using NK Cell Isolation Kit (#130-115-818, Miltenyi Biotec) as per manufacturer's instructions. Approximately $2.5 \times 10^5$ splenic NK cells were transferred via intravenous injection into MC38 tumor bearing C57BL/6 mice 12 days after tumor engraftment, tissues were then collected 3 days later.

## Tumor compartment photo-labeling

Upon reaching 6–8 mm in diameter both subcutaneous and mammary fat pad injected tumors were exposed to a 405-nm wavelength focused LED light (Dymax BlueWave QX4 fitted with 8 mm focusing lens, DYM41572; Intertronics) using 9 cycles of 20 s light exposure with a 5-s break interval between each cycle, at a fixed distance. Black medium density fiberboard was used to shield the remainder of the mouse.

## Tissue dissociation

Tumors were processed as described previously Zhi et al. In short, tumors were cut into small pieces and enzymatically digested using 1 mg/ml Collagenase D (#11088882001, Roche), and 0.1 mg/ml DNase I (#101104159001, Roche) at 37 °C for 20–22 min in a heat block shaker set to 1000 rpm. Samples were then filtered through a 70-µm cell strainer, centrifuged for 5 min at 400 g and 4 °C and resuspended in staining buffer (2%FBS and 2 mM EDTA in PBS).

Spleens were crushed through a 70 µm strainer, then incubated with Gey's red blood cell lysis solution on ice for 5 min. Cells were harvested by centrifuging samples at 400 g for 5 min at 4 °C before resuspending cell pellets in staining buffer. Lymph nodes were collected, cleaned of fat, then cut into little pieces before incubating in

1 mg/ml Collagenase D and 0.05 mg/ml DNase I for 37 °C in a heat block shaker at 1000 rpm for 20 min. Samples were passed through a 70 µm strainer, centrifuged at 400 g and 4 °C for 5 min before resuspending the cell pellet in staining buffer. Livers were cut into small 1–2 mm pieces and pressed through a 100-µm, then washed through a 70 µm strainer with RPMI. Liver suspensions were layered on top a 67% OptiPrep (#07820, STEMCELL Technologies) solution and centrifuged at 1000 g for 25 min before collecting the interphase layer enriched for immune cells. Samples were washed again with RPMI, then centrifuged for 5 min at 400 g and 4 °C before finally resuspending in staining buffer. Lungs were washed, cut into 1–2 mm pieces before incubating tissue in 42.4 µg/ml Liberase (#5401119001, Roche) and 0.02 mg/ml DNase I for 45 min at 37 °C. Samples were passed through a 100-µm, then washed through a 70 µm strainer with RPMI. Cells were pelleted by centrifuging at 400 g and 4 °C for 5 min, then cell pellets were resuspended in Gey's solution and incubated on ice for 5 min. Samples were centrifuged for 5 min at 400 g and 4 °C before resuspending cell pellets in staining buffer. The small intestines were processed as described previously[78], briefly, fat and Peyer's patches were removed for the small intestine, before cleaning internal contents and cutting tissue into small piece in Hank's Balanced Salt Solution (HBSS) (#55037 C, Sigma-Aldrich) containing 2% FBS. Next, the small intestines were shaken vigorously in HBSS with 2 mM EDTA before incubating samples at 37 °C for 20 min, then filtering through a nitex mesh, and washing thoroughly with HBSS. Samples were incubated with 1 mg/ml collagenase VIII (#C2139, Sigma-Aldrich) and 0.1 mg/ml DNase for 15 min, filtered through a 100 µm, and then 70 µm strainer before being centrifuged at 400 g and 4 °C for 5 min, and finally resuspended in staining buffer. Large intestines were processed similarly but digested in 0.85 mg/ml Collagenase V (#C9263, Sigma-Aldrich), 1.25 mg/ml Collagenase D, 1 mg/ml Dispase (#17-105-041, Gibco), and 0.1 mg/ml/ml DNase at 37 °C for 45 min. Large intestines were then processed as per washing and straining steps used for small intestines, and finally cells were resuspended in staining buffer for use. All incubations were done in a shaking incubator set to 300 rpm.

## In vitro *assays*

To assess NK cell mediated target cell killing, B16F0-Ova melanoma tumor cells were co-cultured with FACS NK cells as described previously[79]. B16F10-Ova cells were incubated with 10 µM Cell Trace Violet (#C34557, Invitrogen) for 15 min at 37 °C, washed in RPMI and centrifuged at 350 g for 5 min before plating $5 \times 10^5$ cells/ml in complete RPMI supplemented with 10% FBS and 2 mM L-glutamine. FACS sorted CD11b+ or CD49a+ NK cells from MC38 tumors were added to achieve 1:10, 1:20, and 1:40 Effector: Target ratios. Cells were co-cultured for 4 h at 37 °C and 5% $CO_2$.

To evaluate effect of $PGE_2$ and TGF-β on NK cell phenotype and function, splenic NK cells were MACs enriched and cultured at 37 °C and 5% $CO_2$ in RPMI containing 300U rIL-2 (Peprotech), 50 ng/ml rIL-15 (Peprotech) and either 0.1% DMSO (Sigma) or further supplemented with either 5 ng/ml TGF-β, 43 mg/ml 16,16-Dimethlyprostaglandin E2 (DiPGE2), or both TGF-β and DiPGE2 combined. NK cells were cultured for 48 h before restimulation with PMA and Ionomycin and subsequent flow analysis. In hypoxia experiments, MACs enriched NK cells were cultured with DMSO, TGF-β, or DiPGE2 as above or in a 1% $O_2$ hypoxia chamber in parallel.

## Flow cytometry

To assess cytokine, granzyme and perforin production, cells were stimulated in 50 ng/ml Phorbol 12-myristate 13-acetate (PMA, #P1585, Sigma-Aldrich) and 1.5 µM ionomycin (#I0634, Sigma-Aldrich) for 4 h and in the presence of 10 µg/ml brefeldin A (#B6542, Sigma-Aldrich) added after the first hour. To assess cellular degranulation cells were stimulated like before, but 2 µM Monensin (#420701, BioLegend) was used in place of Brefeldin A, and CD107a was added for the duration of

the stimulation culture. Single cell suspensions underwent Fc blocking with anti-CD16/32 (2.4G2, BioLegend) in staining buffer on ice for 15 min before staining surface antigens in staining buffer on ice for 35 min. Cells were then fixed with BD CytoFix fixation buffer (#554655, BD Biosciences) on ice for 45 min before staining for intracellular markers overnight diluted in eBioscience permeabilization buffer (#00-8333-56, Thermo Fisher Scientific). Secondary intracellular staining to label Granzyme A and CCL5 was performed at room temperature for 45 min. Samples were transferred into staining buffer, with the addition of $1 \times 10^4$ counting beads (#ACBP-100-10, Spherotech) being acquiring data on a BD LSR Fortessa X-20 (BD Bioscience) using FACSDiva 8.0.2 software (BD Bioscience) and analyzed with FlowJo v10 (BD Bioscience). Surface and intracellular antibodies used were against the following mouse antigens: CCL5 purified (1:200, Goat IgG, R&D Systems), CD107a BV786 (1:200, clone 1D4B, BioLegend), CD11b PE-Dazzle594 (1:300, clone M1/70, BioLegend), CD200r1 APC (1:200, clone OX-110, BioLegend), CD3 BV711 or BUV737 (1:200, clone 17A2, BioLegend), CD45 FITC or BV510 (1:200, clone 30-F11, BioLegend), CD49a BUV395 (1:200, clone RM4-5, BD Bioscience), CD49b BV711 (1:100, clone DX5, BioLegend), CD69 PE-Cy7 or BV711 (1:200, clone H1.2F3, BioLegend), DNAM-1 BV605 (1:200, clone TX42.1, BioLegend), EOMES PE or PE-Cy7 (1:100, clone Dan11mag, BioLegend), Granzyme A purified (1:300, clone 3G8.5, BD Biosciences), Granzyme B BV421 (1:300, clone QA18A28, BioLegend), Granzyme C PE-Cy7 (1:300, clone SFC108, BioLegend), IFNγ BUV737 (1:400, clone XMG1.2, BioLegend), IgG2b BV605 or R718 (1:400, clone R12-3, BioLegend), IL-7Rα BV605 or BV421 (1:100, clone A7R34, BioLegend), Ki-67 BV711 or AF700 (1:200, clone SolA15, BioLegend), KLRG1 BV605 (1:100, clone 2F1, BioLegend), LAG3 BV786 (1:100, clone C9B7W, BioLegend), NK1.1 BV650 or BV786 (1:100, clone PK136, BioLegend), NKG2A/C/E BV421 (1:400, clone 20d5, BioLegend), NKp46 BV650 or BV786 (1:100, clone 29A1.4, BioLegend), OX40 BV711 (1:100, clone OX-86, BioLegend), PD-1 PE-Cy7 (1:200, clone RMP1-30, BioLegend), Perforin APC (1:300, clone S16009A, BioLegend), T-bet e660 (1:100, clone eBio4B10, BioLegend), and TCRβ BUV737 (1:200, clone H57-597, BioLegend). AF647-conjugated polyclonal donkey anti-goat IgG (1:400, catalog number #A32849, Thermo Fisher Scientific) was used to identify and amplify CCL5 staining.

Surface and intracellular antibodies used were against the following human antigens: CD16-FITC (1:25, clone 3G8;Biolegend), NKG2A-APC (3:50, clone Z199, Beckman Coulter), GZA-AF700 (1:50, clone CB9, Biolegend), CD49a-BV421 (1:50, clone SR84;BD), CD11b-BV510 (1:25, clone ICRF44; BD), CD3-BV570 (3:50, clone UCHT1; Biolegend), CD57-BV605 (1:50, clone QA17A04; Biolegend), CD103-BV650 (1:25, clone BER-ACT8; BD), PRF1-BV711 (1:50, clone dG9; Biolegend), LAG3-BV786 (1:25, clone T47-530; BD), T-bet-PE (1:10, clone O4-46;BD), GZB-PE-CF594 (1:25, clone GB11; BD), CD127-PE-Cy5 (3:50, clone A019D5; Biolegend), CD69-PE-Cy5.5 (1:25, clone CH/4; Thermofisher), EOMES-PE-Cy7 (3:50, clone WD1928; Thermofisher), CD45-BUV395 (1:25, clone HI30; BD), CD56-BUV737 (1:25, clone NCAM16.2; BD).

## FACS isolation
Single cell suspensions were stained for antibodies raised against the following antigens: CD45 FITC (clone 30-F11, BioLegend), CD11b PE-Dazzle594 (clone M1/70, BioLegend), TCRβ BUV737 (clone H57-597, BioLegend), NKp46 BV650 (clone 29A1.4, BioLegend), and CD49a BUV395 (clone RM4-5, BD Biosciences), in addition to Live/Dead Near-IR viability stain (#L10119, Thermo Fisher Scientific). NK cells (Live CD45$^+$ CD11b$^{-/low}$, TCRβ$^-$ CD3$^-$ NKp46$^+$) were sorted into CD11b$^+$ CD49a$^-$ and CD11b$^-$ CD49a$^+$ cells using a FACS Aria II Cell sorter (BD Biosciences) and then used for in vitro assays. For scRNA-seq, singles cells were stained using antibodies against the following: CD45-BUV395 (clone 30-F11, BioLegend), CD11b-APC (clone M1/70, BioLegend), Ter119-PE-Cy7 (1:400, clone TER-119, BioLegend), NK1.1-BV650 (clone PK136, BioLegend). Tumor infiltrating lymphocytes (Live CD45$^+$ Ter119$^-$ CD11b$^{-/low}$) were

sorted into two groups based on the presence of the Kaede Red, namely, G48 and R48 referring to Kaede Green 48 h and Kaede Red 48 h.

## Single-cell library construction and sequencing
Gene expression libraries from were prepared from FACS-sorted populations of single cells using the Chromium Controller and Chromium Single Cell 3' GEM Reagent Kits v3 (10x genomics, Inc.) according to the manufacturer's protocol. The resulting sequencing libraries comprised of standard Illumina paired-end constructs flanked with P5 and P7 sequences. The 16 bp 10x barcode and 10 bp UMI were encoded in read 1, while read 2 was used to sequence the cDNA fragment. Sample index sequences were incorporated as the i7 index read. Paired-end sequencing ($2 \times 150$ bp) was performed on the Illumina NovaSeq 6000 platform. The resulting.bcl sequence data were processed for QC purposes using bcl2fastq software (v2.20.0.422) and the resulting fastq files were assessed using FastQC (v0.11.3), FastqScreen (v0.9.2) and FastqStrand (v0.0.5) prior to alignment and processing with the CellRanger (v6.1.2) pipeline.

## Processing of scRNA-seq
Single-cell gene expression data from CellRanger count output (filtered features, barcodes, and matrices) were analyzed using the Scanpy[80] (v1.8.2) workflow. Doublet detection was performed using Scrublet[81] (v0.2.1), with cells from iterative sub-clustering flagged with outlier Scrublet scores labeled as potential doublets. Cells with counts mapped to >6000 or <1000 genes were filtered. The percentage mitochondrial content cut-off was set at <7.5%. Genes detected in fewer than 3 cells were filtered. Total gene counts for each cell were normalized to a target sum of $10^4$ and log1p transformed. There resulted in a working dataset of 46,342 cells. Next, highly variable features were selected based on a minimum and maximum mean expression of ≥0.0125 and ≤3 respectively, with a minimum dispersion of 0.5. Total feature counts, mitochondrial percentage, and cell cycle scores, where indicated, were regressed out. The number of principal components used for neighborhood graph construction was set to 50 initially, and subsequently 30 for subgroup processing. Clustering was performed using the Leiden algorithm with resolution set between 0.8 and 1.0. Uniform manifold approximation and projection (UMAP, v0.5.1) was used for dimensional reduction and visualization, with a minimum distance of 0.3, and all other parameters according to the default settings in Scanpy.

## Analysis of scRNA-seq from mouse tumor models
Cell types of interest were subset and re-clustered as described above. Resulting clusters were annotated using canonical marker genes. Gene set scoring was performed using Scanpy's tl.score genes tool. Gene sets were obtained from the Molecular Signature Database (MSigDB) inventory, specifically KEGG or Gene Ontology (GO), using the R package msigdbr (v7.5.1) or published RNAseq data. Differential gene testing was performed using the Wilcoxon rank sum test with Benjamini-Hochberg multiple-testing correction implemented in Scanpy's tl.rank_genes_groups. Trajectory analysis was performed using partition-based graph abstraction (PAGA)[82] and diffusion pseudotime[83] (implemented in Scanpy v1.8.2). For pseudotime analysis, the differentiation trajectory was rooted in the Kaede-green dominant cluster (NK_1), which represents the cellular subset most associated with newly entering cells. The specific root cell was selected based on the extrema of diffusion components. Calculation of differential expression across pseudotime was performed using the tradeSeq (v0.99) package[39]. The 24 and 72 h post-photoconversion scRNA-seq data was processed and analysed following the same steps above; and the data is accessible through the ArrayExpress repository (accession number E-MTAB-10176), as previously published and described[34]. Integration and label transfer of the 24/72 h scRNA-seq data, cycling NK cells, and PBMCs (obtained from 10X Genomics

demonstration data: https://www.10xgenomics.com/resources/datasets/10-k-mouse-pbm-cs-multiplexed-2-cm-os-3-1-standard-6-0-0) with the main data set was performed using Scanpy's tl.ingest tool. For cycling NK cells, cell cycle regression was first performed on subset *Mki67*[+] cells using Scanpy's pp.regress_out function, before re-integration. Transcription factor activity enrichment analysis was performed using pySCENIC (v0.12)[40]. Cell-cell communication analysis was performed using CellChat (v1.5.0)[45].

## Analysis of RNA-seq from human tumors

Published human CRC scRNA-seq data and accompanying metadata was accessed and downloaded from the GEO repository using the ascension number GSE178341[53]. The cancer genome atlas (TCGA) bulk-RNAseq data of human colorectal adenocarcinoma (TCGA-COAD) was accessed using TCGAbiolinks[52]. Cellular deconvolution was performed using the Cibersortx webtool[84]. scRNA-seq from human colorectal tumors was analyzed using the Scanpy (v1.8.2) workflow as outlined above. Filtering for quality control was performed according to the parameters outlined in the original publication. Batch correction was performed using the Harmony algorithm (harmonypy v0.0.5) with patient ID as the batch term. Cell annotations from original publications were checked and refined using canonical marker gene expression. Gaussian kernel density estimation to compute density of cells from various conditions in the UMAP embedding was performed using Scanpy's tl.embedding_density. Pre-ranked gene set enrichment analysis (GSEA) was implemented in fgsea (v1.24), using the Wald statistic as the gene rank metric.

## Statistical analysis

Data collected were analyzed using Flow Jo 10.8.1 software (BD Bioscience) and GraphPad Prism 9.4.0. UMAPs generated in FlowJo were made using DownSampleV3 and UMAP plugins, both available on the FlowJo Exchange. scRNA-seq data was analysed in Python (v3.8.12) and R (v4.0.4) Normality was determined using the Shapiro-Wilk test. Pairs of samples were compared using an unpaired two-tailed Mann–Whitney U test. When comparing more than two sets of data, statistical significance was determined by either one-way ANOVA with Tukey's multiple comparison test, or two-way ANOVA with Šidák's multiple comparisons test. Two data points were identified as outliers by the ROUT method in Supplemental figure 11E-F and removed from analysis. All graphs show mean ± SD unless stated otherwise. Only significant differences are marked on graphs.

## Reporting summary

Further information on research design is available in the Nature Portfolio Reporting Summary linked to this article.

# Data availability

The scRNA-seq data generated by this study have been deposited in the GEO public repository under accession numbers GSE221064.

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

## Acknowledgements

We thank Dr Y. Miwa (Tsukuba University) and Dr O. Kanagawa (RCAI, RIKEN) and Dr. M. Tomura (Osaka Ohtani University) for the Kaede mice. We thank Dr S. Zelenay for kindly sharing MC38$^{ptgs-/-}$ cells. We thank the Biomedical Services Unit at the University of Birmingham for all their help with in vivo experimental work. We thank A. Ptasinska and colleagues at Genomics Birmingham, the genomic and sequencing facility of the University of Birmingham, for their help with single cell RNA sequencing and the University of Birmingham Flow Cytometry Platform. We thank Gareth Howell and the University of Manchester flow cytometry core, and Andy Hayes and Claire Morrisroe in the University of Manchester Genomic Technologies core facility for their help with single cell RNA sequencing. Results shown here are in part based upon data generated by the TCGA Research Network: https://www.cancer.gov/tcga. This work was supported by the following Grants to D.R.W.: Senior Research Fellowship from the Wellcome Trust (110199/Z/15/Z), Cancer Research UK Immunology Project Award (C54019/A27535), Cancer Research Institute CLIP Grant (CRI3128), Worldwide Cancer Research Grant (21-0073) and an MRC IMPACT iCASE Studentship with AstraZeneca. Research in the laboratory of J.M. was supported by The Swedish Cancer Society. Funding to M.R.H.: Sir Henry Dale Fellowship jointly funded by the Wellcome Trust and the Royal Society (Grant Number 105644/Z/14/Z), and Lister Institute of Preventative Medicine Prize. Z.K.T. and M.R.C. are supported by a Medical Research Council Human Cell Atlas Research Grant (MR/S035842/1). M.R.C. is supported by a Wellcome Trust Investigator Award (220268/Z/20/Z) and by the NIHR Cambridge Biomedical Research Centre.

## Author contributions

I.D. designed and performed experiments, analyzed data, performed statistical analysis and wrote the manuscript; C.Y.C.L. designed and performed experiments, analyzed data and wrote the manuscript; Z.K.T., Z.L., C.W., F.G., B.C.K., V.M.R., R.F. designed and performed experiments; C.A.T. designed and performed experiments and analyzed data; S.M.B. analyzed data; C.N. and U.L. obtained human tissue samples; C.S., V.S. provided in vivo model and critiqued data; G.C., S.A.H., S.J.D., J.M., M.R.H. designed experiments and critiqued data; M.R.C. and D.R.W. designed experiments, analyzed and critiqued data and wrote the manuscript.

## Competing interests

The authors declare the following competing interests: G.C., S.A.H., S.J.D. are full-time employees of AstraZeneca and own or have owned AstraZeneca stock. The remaining author declare no competing interests.
