## [Peer Review File · Nature Communications]

Rapid functional impairment of natural killer cells following tumor entry limits anti-tumor immunityREVIEWER COMMENTS

Reviewer #1 (expert in NK cells and innate lymphoid cells):

In this work, Dean et al. investigated the fate of NK cells after tumor invasion using a photo-labeling mouse model. They also investigated how to resuscitate NK cell functions in the tumor. The study largely replicates data Gao et al. published in Nature Immunology in 2017. The real 'novelty' is in the part that revives NK cells.

Apart from the lack of novelty, there is a real problem with cell identity in scRNAseq analysis, as the gene signatures of NK and ILC1 described in the literature and cited by the authors (Criler et al; Lopes et al.; McFarland et al.;...) are not used to uniquely identify these subsets (NK and ILC1).

Moreover, Ncr1 is not expressed by NKT cells, for example.

Reviewer #2 (expert in single-cell transcriptomics and genomics):

Using Kaede photoconvertible mouse combined with murine cancer model, the authors have traced the phenotypic change of NK cells in cancer over time, reporting the rapid transition from the cytotoxic, Cd11b high status to Cd49a high status with reduced activity. Also, the authors confirm that the tumor resident NK cells have a neglectable effect on tumor growth control, while this can be reversed by stimulating them with IL15 pathway. The authors also identified the existence of ITGA1 high, less cytotoxic population of NK cells in human CRC, which are enriched in advanced stages. The study is well-designed and the manuscript is easy to follow with clear conclusions and rich experimental support. The conclusion has an important contribution to the understanding of the immune environments of tumors. At the same time, I would like to add some suggestions for the further improvements as follows:

1. Further insights on the mechanism of NK cell conversion

The authors have a beautiful pseudotime trajectory for NK cell conversion. However, the mechanism for the NK cell conversion in cancer tissue is not thoroughly investigated yet. I would suggest the authors perform an analysis of upstream regulators for the transition by applying tools like SCENIC or PROGENy and provide explanations on potential regulatory mechanisms.

2. Adding more context information

It would be great if the authors could compare the NK cells found in cancer tissue with blood NK cells by integrating the public dataset to show provide information on the source of NK cells in tumors, considering the heterogeneity of NK cells in the blood. Also, it would be great to do the same analysis by comparing human PBMC and additional cancer types. This could generalize the phenotypic definition of tumor-resident NK cells and their sources.

3. Cross-species comparison

In addition to the marker-based approaches to finding murine-defined NK cell counterparts in human CRC, it would be ideal to perform DEG analysis between Cd11b vs Cd49a population between mouse and human and show the overlap between DEGs by van diagram or scatterplots.

4. Longevity and function of tumor-adapted NK cells

For the NK cells who lost their activity in tumors, what would be their functions? Are they just filling up the space? How long do they stay in the cancer? Do they form any spatial

localization patterns? Could the authors provide discussions on this? They could also perform CellChat analysis to show how these two populations differ in their interaction between cells constituting tumor microenvironments.

Reviewer #3 (expert in NK cells and tumour immunology):

Dean and colleagues describe the NK cells infiltrating solid tumors by using photo-labeling and longitudinal transcriptomic analysis and report that IL-15 can restore function to tumor-infiltrating NK cells in mouse tumor models. The study is well performed, informative, and properly controlled. The photo-labeling provides new insights into the rapid acquisition of CD49a on NK cell entering tumors, their lack of egress, and loss of effector functions. Overall, the manuscript is excellent. As the data are already available, it would be worthwhile for the authors to address whether the cycling NK cells in the tumors have a unique phenotype and if there is evidence for clonal expansion of the resident cells based on skewing of the Ly49 transcripts in the scRNA-Seq data set.

Dean et al: Response to reviewer's comments

We thank all the Reviewers for their comments and suggestions. We have responded to each below. Collectively, these revisions (2 new Supplementary Figures, further panels added to 2 existing Supplementary Figures and a Supplementary Table) have provided important clarification of data within our manuscript. We have also re-written much of the Discussion, to streamline the text, focus on the key findings and bring in aspects of the requested revisions.

Changes to text in the manuscript have been highlighted in yellow. Requested analyses and data are shown below and where we have included them within the revised manuscript, we have further signposted this (in italics).

Reviewer #1 (expert in NK cells and innate lymphoid cells):

In this work, Dean et al. investigated the fate of NK cells after tumor invasion using a photo-labeling mouse model. They also investigated how to resuscitate NK cell functions in the tumor. The study largely replicates data Gao et al. published in Nature Immunology in 2017. The real 'novelty' is in the part that revives NK cells.

Apart from the lack of novelty, there is a real problem with cell identity in scRNAseq analysis, as the gene signatures of NK and ILC1 described in the literature and cited by the authors (Criler et al; Lopes et al.; McFarland et al.;...) are not used to uniquely identify these subsets (NK and ILC1). Moreover, *Ncr1* is not expressed by NKT cells, for example.

We thank the Reviewer for their comments and time spent considering our manuscript. We agree that our study builds on the Gao *et al* manuscript as well as the Cortez *et al* manuscript published together in Nature Immunology. We would politely disagree that our data only replicates the information in these initial studies. Utilising our dynamic labelling approaches our data provides new insight into the rate at which NK cells adapt to the tumour environment and importantly, the fate of these cells over time within this tissue. Our study further provides extensive insight into the composition of the NK compartment in tumours and how this is established. Collectively we think this expands upon and clarifies the initial observations published in Nature Immunology. We also anticipate that our sequencing based analyses will provide useful resources for the field.

We thank the reviewer for their suggestion to perform signature-based annotation of the NK/ILC clusters, in addition to the marker-based annotation presented in the manuscript. We have done the requested analysis using NK/ILC-1 specific signatures from the referenced manuscripts. *Of note, the NK / ILC1 signatures in Crinier et al. (2018) were from Robinette et al. 2015.* The signature enrichment scores clearly support the current annotations in our scRNA-seq data (shown below), alongside our existing data such as the absence of *Eomes* expression in the ILC cluster (already evident in the original Supplementary Figure 1).

We have included the requested new data (below) showing the signature enrichment within Supplementary Figure 1 (Fig. S1J), as we agree with the Reviewer that this comparison provides further support for our cluster identification. Text describing this new data is on Page 6.

Regarding our NKT annotation, *Ncr1* expression is low in the NKT cluster (Fig 1B), certainly lower than in other NK cell clusters. However, *Ncr1* expression, including by scRNA-seq, has been described in NKT cells (for example PMID 21364281, PMID 33173202) indicating that some NKT subsets, or NKT cells under certain conditions, do express *Ncr1*. More importantly, as shown in the manuscript, the NKT cluster express NKT cell-associated transcripts, including TCR transcripts (*Trac*, *Trbc1*, *Trbc2*), *Cd3d*, *Cd3e*, and *Lef1*. We have shown this data below for clarity (Rebuttal Fig. 1) We did not think it necessary to include this data in the manuscript given the existing data already within Fig. S1. Moreover, in all flow cytometry experiments, antibody staining of CD3 has been used to exclude T cells, so all subsequent analysis have excluded NKT cells.

Rebuttal Fig. 1 – NKT-associated transcripts across the ‘NK’ clusters

Reviewer #2 (expert in single-cell transcriptomics and genomics):

Using Kaede photoconvertible mouse combined with murine cancer model, the authors have traced the phenotypic change of NK cells in cancer over time, reporting the rapid transition from the cytotoxic, Cd11b high status to Cd49a high status with reduced activity. Also, the authors confirm that the tumor resident NK cells have a neglectable effect on tumor growth control, while this can be reversed by stimulating them with IL15 pathway. The authors also identified the existence of ITGA1 high, less cytotoxic population of NK cells in human CRC, which are enriched in advanced stages. The study is well-designed and the manuscript is easy to follow with clear conclusions and rich experimental support. The conclusion has an important contribution to the understanding of the immune environments of tumors. At the same time, I would like to add some suggestions for the further improvements as follows:

We thank the Reviewer for their supportive comments.

1. Further insights on the mechanism of NK cell conversion

The authors have a beautiful pseudotime trajectory for NK cell conversion. However, the mechanism for the NK cell conversion in cancer tissue is not thoroughly investigated yet. I would suggest the authors perform an analysis of upstream regulators for the transition by applying tools like SCENIC

or PROGENy and provide explanations on potential regulatory mechanisms.

We thank the reviewer for their suggestion to explore potential transcriptional regulators of the proposed NK trajectory and agree that such an analysis would provide some initial indications of the potential mechanisms involved and likely serve as a useful resource for the field. Since transcription factor expression may not always be detected by scRNA-seq, and transcription factor expression may not reflect activity per se, we performed the suggested gene regulatory network analysis using SCENIC. This analysis revealed the concomitant downregulation of a large number of transcription factor regulons over pseudotime, as NK cells acquire the tumour-resident phenotype. Many of these have been described to play crucial roles in NK cell activation, cytotoxicity and cytokine production. We have briefly highlighted a small number of these (conscious of space requirements and attaching too much weight to this initial analysis). We further note that we did not identify transcription factor regulons significantly upregulated with pseudotime.

These data are now show in a new Supplementary Figure (Fig. S2) with text describing this data on Page 7 of the Results section and also some further text within the Discussion.

2. Adding more context information

It would be great if the authors could compare the NK cells found in cancer tissue with blood NK cells by integrating the public dataset to show provide information on the source of NK cells in tumors, considering the heterogeneity of NK cells in the blood. Also, it would be great to do the same analysis by comparing human PBMC and additional cancer types. This could generalize the phenotypic definition of tumor-resident NK cells and their sources.

We thank the Reviewer for this suggestion. By integrating sorted Kaede-green NK cells 24h post-photoconversion (**Fig 1L**), we have proposed that most newly-entered NK cells are NK_1 in our model, consistent with the computed pseudotime trajectory. It is likely that these NK cells are entering the tumour from the blood. To specifically address the Reivewer's comment, we have additionally analysed NK cells from mouse PBMCs (using a public scRNA-seq dataset: <https://www.10xgenomics.com/resources/datasets/10-k-mouse-pbm-cs-multiplexed-2-cm-os-3-1-standard-6-0-0>), to compare with tumour NK cells. Integration and label transfer of blood NK cells demonstrated that these most resembled NK_1 and NK_2, which are predominantly Kaede-green in the scRNA-seq data, and importantly, none were mapped to NK_5, the tumour-retained NK cell state. These new data are shown below alongside further comparison of *Itgam*, *Itga1* and *Itga2* expression, which further indicated that blood NK cells were most similar to NK_1 (and importantly, no *Itga1*

expression was detected in the PBMC NK cells. These data support our conclusions that NK cells enter the tumour and rapidly acquire a distinct tumour-retained state not present in the periphery.

We have included these data (below) in Supplementary Figure 1 (Fig. S1M-O) and the accompanying text is on Pages 7-8.

Regarding the requested comparison of human NK cells in peripheral blood versus different cancers, this is obviously a substantial bioinformatics undertaking. An extensive manuscript on exactly this topic was recently published in Cell (Tang et al – ‘A pan-cancer single-cell panorama of human natural killer cells’). Given this publication and the scale of a further analysis (estimated to take many months of dedicated bioinformatics time), we would politely suggest that this is beyond the scope of our manuscript and already now available to researchers in the field. We have cited the Tang et al. manuscript on Page 15.

3. Cross-species comparison

In addition to the marker-based approaches to finding murine-defined NK cell counterparts in human CRC, it would be ideal to perform DEG analysis between Cd11b vs Cd49a population between mouse and human and show the overlap between DEGs by van diagram or scatterplots.

We thank the reviewer for the suggestion and have performed the analysis requested, specifically, comparing the DEGs between *Itgam*-expressing and *Itga1*-expressing tumour NK cells in human and murine CRC (data from Fig 1A and Fig 5A in our manuscript). This data is shown below. A large number of DEGs overlapped, indicating that similar NK cell states exist across species, and may represent the acquisition of a common tumour-resident phenotype. Of note, there were many more DEGs among murine NK cells because the larger number of cells and FACS-sorting of Kaede-green/red populations enabled us to contrast NK cell states at more polar ends of the spectrum/transition (NK_1 vs NK_5) compared to in the human CRC data (NK1-1 vs NK1-3).

We have included this analysis (below) in Supplementary Figure 11 (Fig. S11G) and text describing this new data is on Page 14. A Supplementary Table of the DEGs is also included.

4. Longevity and function of tumor-adapted NK cells

For the NK cells who lost their activity in tumors, what would be their functions? Are they just filling up the space? How long do they stay in the cancer? Do they form any spatial localization patterns? Could the authors provide discussions on this? They could also perform CellChat analysis to show how these two populations differ in their interaction between cells constituting tumor microenvironments.

We thank the Reviewer for this suggestion. Firstly we have performed the cell-cell communication analysis suggested by the reviewer, using CellChat. This analysis demonstrated that tumour-retained NK_5 undertake very different interactions with tumour-infiltrating immune cells, compared to NK cells earlier in the trajectory, as one might expect (see Rebuttal Fig. 2 below). Unbiased clustering of cell-cell communication patterns reveal that NK_5 have a distinct pattern of interaction from NK_1-4 (pattern 6 vs pattern 1). Of note, pattern 6 is also shared by ILC1. These data indicate that tumour-retained NK interact with TME immune cells in a profoundly different manner. (Note this was done with the full CD45+ scRNA-seq data we have). We have included this requested analysis here only as a rebuttal figure, as we were conscious to avoid over-crowding our manuscript with vague speculation regarding the interactions NK cells make in the tumour.

Rebuttal Fig. 2 – Overview of the cellular interactions within the TME

But we agree with the reviewer that evidence linking the different NK states we have defined in the tumour with distinct cellular interactions is informative and of interest to the field. To try to focus the analysis into some specific interactions of potential relevance, we analysed NK cell – myeloid cell interactions only (excluding mast cells, LC, pDC etc. where $n < 100$). This analysis revealed a number of potential interactions, some of which (e.g. *Ifng*, *Ccl5*) are further developed within the manuscript and provided an unbiased means of introducing potential changes in NK cell function over time in the tumour. Therefore we considered that this analysis illustrated the point we felt the Reviewer was trying to make and have included this in the manuscript as a new Supplementary Figure.

This NK-myeloid interaction analysis (below) is included as a new Supplementary Figure (Fig. S7) and these data are briefly described on Page 10. As also requested by the Reviewer we added some further text considering the retained NK population to the Discussion.

Reviewer #3 (expert in NK cells and tumour immunology):

Dean and colleagues describe the NK cells infiltrating solid tumors by using photo-labeling and longitudinal transcriptomic analysis and report that IL-15 can restore function to tumor-infiltrating NK cells in mouse tumor models. The study is well performed, informative, and properly controlled. The photo-labeling provides new insights into the rapid acquisition of CD49a on NK cell entering tumors, their lack of egress, and loss of effector functions. Overall, the manuscript is excellent. As the data are already available, it would be worthwhile for the authors to address whether the cycling NK cells in the tumors have a unique phenotype and if there is evidence for clonal expansion of the resident cells based on skewing of the *Ly49* transcripts in the scRNA-Seq data set.

We thank the reviewer for the helpful suggestions. We agree that for improved clarity of the data we should address whether cycling NK cells have a unique phenotype. Therefore, we have sub-setted all *Mki67+* cells from the scRNA_seq data, and regressed out cell cycle genes to eliminate these transcripts as a source of variation. We then re-integrated the *Mki67+* cells with *Mki67-* tumour NK cells and performed a label transfer to annotate them. This analysis demonstrated that cycling cells could be found in all NK / ILC / NKT cell clusters, indicating that the cycling NK cells in our scRNA-seq

data do not constitute a distinct phenotype. However, it was notable that NK_5 was particularly enriched among the cycling NK cells, suggesting that some tumour-retained NK cells may expand *in situ*.

This data (below) is now included in Supplementary Figure 1 (**Fig. S1K, L**) and accompanying text is on Page 6.

*Chi-squared test for NK_5 was used.

We have performed an analysis of the Ly49 transcripts as requested by the Reviewer. These data are shown below in Rebuttal Fig. 3. We have not included these data as we were not convinced that they really added to the (already data packed) manuscript.

Rebuttal Fig. 3 – Ly49 transcripts across the ‘NK’ clusters

REVIEWERS' COMMENTS

Reviewer #1 (expert in NK cells and innate lymphoid cells):

the reviewer thanks the authors for their additional analysis of the data on the identity of the cells examined and supports the publication of this study.

Reviewer #2 (expert in single-cell transcriptomics and genomics):

The authors have adequately addressed my points. I'm happy with the publication of the revised version.

Reviewer #3 (expert in NK cells and tumour immunology):

Revisions are acceptable.

Dean et al. Response to Reviewers Comments after resubmission of revised manuscript

The Reviewers' comments are shown in full below. There are no further comments to address. We again thank the Reviewers for their support of our manuscript

REVIEWERS' COMMENTS

1. Reviewer #1 (expert in NK cells and innate lymphoid cells):

the reviewer thanks the authors for their additional analysis of the data on the identity of the cells examined and supports the publication of this study.

2. Reviewer #2 (expert in single-cell transcriptomics and genomics):

The authors have adequately addressed my points. I'm happy with the publication of the revised version.

3. Reviewer #3 (expert in NK cells and tumour immunology):

Revisions are acceptable.